# ZEB1-mediated fibroblast polarization controls inflammation and sensitivity to immunotherapy in colorectal cancer

Constantin Menche [1,2,14], Harald Schuhwerk [3,14], Isabell Armstark[3], Pooja Gupta[4], Kathrin Fuchs[3], Ruthger van Roey[3], Mohammed H Mosa[1,2], Anne Hartebrodt[5], Yussuf Hajjaj[3], Ana Clavel Ezquerra [3], Manoj K Selvaraju[4], Carol I Geppert[6], Stefanie Bärthel [7,8,9], Dieter Saur [7,8,9,10], Florian R Greten [1,2,11,12], Simone Brabletz [3], David B Blumenthal [5], Andreas Weigert [2,12,13], Thomas Brabletz [3✉], Henner F Farin [1,2,11,12,15✉] & Marc P Stemmler [3,15✉]

## Abstract

The EMT-transcription factor ZEB1 is heterogeneously expressed in tumor cells and in cancer-associated fibroblasts (CAFs) in colorectal cancer (CRC). While ZEB1 in tumor cells regulates metastasis and therapy resistance, its role in CAFs is largely unknown. Combining fibroblast-specific *Zeb1* deletion with immunocompetent mouse models of CRC, we observe that inflammation-driven tumorigenesis is accelerated, whereas invasion and metastasis in sporadic cancers are reduced. Single-cell transcriptomics, histological characterization, and in vitro modeling reveal a crucial role of ZEB1 in CAF polarization, promoting myofibroblastic features by restricting inflammatory activation. *Zeb1* deficiency impairs collagen deposition and CAF barrier function but increases NFκB-mediated cytokine production, jointly promoting lymphocyte recruitment and immune checkpoint activation. Strikingly, the *Zeb1*-deficient CAF repertoire sensitizes to immune checkpoint inhibition, offering a therapeutic opportunity of targeting ZEB1 in CAFs and its usage as a prognostic biomarker. Collectively, we demonstrate that ZEB1-dependent plasticity of CAFs suppresses anti-tumor immunity and promotes metastasis.

**Keywords** Colorectal Cancer; Tumor Microenvironment; Cancer-Associated Fibroblast Plasticity; Immune Checkpoint Blockade
**Subject Categories** Cancer; Immunology; Signal Transduction

## Introduction

Colorectal cancer (CRC) is the third most frequent tumor type that accounts for the second highest cancer-related mortality worldwide (Morgan et al, 2023). Despite improved diagnosis and treatment options at an early stage, advanced CRC frequently leads to fatal metastatic relapse. Molecular classification has shown that the stroma-rich consensus molecular subtype 4 (CMS4) is linked with the worst prognosis in patients (Guinney et al, 2015). A refinement of this classification based on recent single-cell RNA sequencing (scRNA-seq) demonstrated that fibroblast enrichment together with tumor cell-intrinsic features contribute to poor prognosis (Joanito et al, 2022). These analyses identify cancer-associated fibroblasts (CAFs) as crucial players in the tumor microenvironment (TME) that drive disease progression, therapy resistance, and metastasis (Calon et al, 2015; Isella et al, 2015; Sahai et al, 2020; Schmitt and Greten, 2021).

Immune checkpoint blockade (ICB) therapy has shown impressive efficacy for patients with microsatellite instable (MSI) tumors (Andre et al, 2020; Le et al, 2015). These tumors display an elevated tumor mutation burden resulting in high immunogenicity. However, for most patients with microsatellite stable (MSS) tumors, no immunotherapies are available. In preclinical mouse models, inhibition of TGFβ signaling in CAFs promote lymphocyte infiltration and response to ICB (Tauriello et al, 2018). This example highlights the concept of a stroma-directed intervention to improve therapy of MSS tumors. However, the identification of specific targets to alter the fibroblast-rich stroma is needed.

[1]Georg-Speyer-Haus, Institute for Tumor Biology and Experimental Therapy, Frankfurt am Main, Germany. [2]Frankfurt Cancer Institute, Goethe University Frankfurt, Frankfurt/Main, Germany. [3]Department of Experimental Medicine 1, Nikolaus-Fiebiger Center for Molecular Medicine, FAU Erlangen-Nürnberg, Erlangen, Germany. [4]Core Unit for Bioinformatics, Data Integration and Analysis, Center for Medical Information and Communication Technology, University Hospital Erlangen, FAU Erlangen-Nürnberg, Erlangen, Germany. [5]Biomedical Network Science Lab, Department Artificial Intelligence in Biomedical Engineering (AIBE), FAU Erlangen-Nürnberg, Erlangen, Germany. [6]Institute of Pathology, University Hospital Erlangen, FAU Erlangen-Nürnberg, Erlangen, Germany. [7]Division of Translational Cancer Research, German Cancer Research Center (DKFZ) and German Cancer Consortium (DKTK), Heidelberg, Germany. [8]Chair of Translational Cancer Research and Institute of Experimental Cancer Therapy, Klinikum rechts der Isar, School of Medicine, Technische Universität München, Munich, Germany. [9]Center for Translational Cancer Research (TranslaTUM), School of Medicine, Technical University of Munich, Munich, Germany. [10]Department of Internal Medicine II, Klinikum rechts der Isar, Technische Universität München, Munich, Germany. [11]German Research Center (DKFZ), Heidelberg, Germany. [12]German Cancer Consortium (DKTK), Partner Site Frankfurt/Mainz, Frankfurt am Main, Germany. [13]Institute of Biochemistry I, Goethe University, Frankfurt am Main, Germany. [14]These authors contributed equally to this work as first authors: Constantin Menche, Harald Schuhwerk. [15]These authors contributed equally to this work as senior authors: Henner F Farin, Marc P Stemmler. ✉E-mail: thomas.brabletz@fau.de; h.farin@georg-speyer-haus.de; marc.stemmler@fau.de

Recent scRNA-seq analyses have shown that CAFs represent a heterogeneous cell population with several prototypic subtypes as initially identified in PDAC and later in other solid cancers including CRC (Bartoschek et al, 2018; Biffi et al, 2019; Elyada et al, 2019; Kieffer et al, 2020; Lee et al, 2020; Ohlund et al, 2017; Sebastian et al, 2020). Myofibroblast-like myCAFs localize closely to tumor cells and secrete extracellular matrix (ECM) encapsulating tumor cells, whereas iCAFs in the tumor periphery create an inflammatory milieu (Ohlund et al, 2017; Sahai et al, 2020). In addition, MHCII+ antigen-presenting CAFs (apCAFs) have been described (Elyada et al, 2019). The importance of CAF plasticity has been recently illustrated in rectal cancer, where IL1-induced iCAFs adopted a senescence phenotype, which mediates radioresistance and disease progression. Therapeutic inhibition of IL1 signaling allowed to restore radiosensitivity, indicating iCAFs as promising therapeutic target (Nicolas et al, 2022).

ZEB1 is a core EMT-transcription factor and is frequently upregulated at the invasive front of CRC and other cancer entities, where it orchestrates tumor stemness, metastasis, and therapy resistance (Caramel et al, 2018; Stemmler et al, 2019). Its tumor cell-specific loss leads to profound changes in gene expression, impairing cell plasticity (Krebs et al, 2017; Meidhof et al, 2015; Preca et al, 2015; Wellner et al, 2009). In pancreatic and breast cancer, ZEB1 was also found to be upregulated in the dysplastic fibroblast-rich stroma, which was correlated with poor survival (Bronsert et al, 2014; Fu et al, 2019). However, the functional relevance of ZEB1 in CAFs and its effect on tumor progression is yet unknown. Here, we have studied the role of ZEB1 in fibroblasts using mouse models for colitis-associated cancer and sporadic metastatic CRC demonstrating a key requirement for CAF diversification. Analysis of the immune TME identified a stage-specific effect on lymphocyte infiltration and CRC progression. $Zeb1$ loss reduced liver metastasis and augmented responsiveness to ICB, highlighting the potential of a CAF-directed therapy.

# Results

## ZEB1 in fibroblasts affects CRC tumorigenesis in a context-dependent manner

ZEB1 is expressed heterogeneously in the TME of both human and murine tumors (Fig. 1A,B), and high expression of ZEB1 in stromal cells of pancreatic cancer is a prognostic factor for poor survival in patients (Bronsert et al, 2014). To investigate a functional involvement, we generated fibroblast-specific $Zeb1$-deleted mice (Fib$^{\Delta Zeb1}$), by combining the $Zeb1^{fl/fl}$ genotype (Brabletz et al, 2017) with the constitutive Col6a1-Cre transgene (Koliaraki et al, 2015) or with the tamoxifen-inducible Col1a2-CreERT2 knock-in allele (Zheng et al, 2002). Cre recombinase activity in both lines was restricted to fibroblasts as identified by Cre reporter alleles (Appendix Fig. S1A,B), and deletion of $Zeb1$ was confirmed at the genomic (Appendix Fig. S1C) and protein level in the colon of both models (Appendix Fig. S1D,E). Of note, loss of $Zeb1$ in fibroblasts did not compromise colon morphogenesis or homeostasis (Appendix Fig. S1F,G).

Fib$^{\Delta Zeb1}$ mice were subjected to the inflammation-driven AOM/DSS model (Neufert et al, 2007) (Fig. 1C), where deletion in fibroblasts did not affect overall or tumor-free survival of mice

(Appendix Fig. S2A,B). Intriguingly, endoscopic scoring revealed increased colonic obstruction (Fig. 1D), which was confirmed by significantly increased individual and total adenoma volumes and a trend to increased tumor numbers in Fib$^{\Delta Zeb1}$ mice at endpoints (Fig. 1E,F; Appendix Fig. S2C). Histologic differentiation was not affected by loss of stromal $Zeb1$ (Appendix Fig. S2D), but endoscopic analysis showed slightly increased colonic inflammation (Appendix Fig. S2E–G). By immunohistochemistry (IHC) fewer ZEB1-positive stromal cells and increased epithelial proliferation accompanied by slightly reduced cell death was observed in Fib$^{\Delta Zeb1}$ mice (Fig. 1G,H). These data demonstrate that loss of $Zeb1$ in fibroblast promotes inflammation-driven adenoma growth.

Sporadic CRC progression was modeled by orthotopic transplantation of tumor organoids (Fumagalli et al, 2018) into Fib$^{\Delta Zeb1}$ and Fib$^{Ctrl}$ mice (Fig. 1I). Tumor organoids were genetically engineered from normal colonic organoids to harbor mutations in $Apc$, $Tp53$ and $Kras$ loci ($Apc^{\Delta/\Delta}$, $Kras^{G12D}$, $Tp53^{\Delta/\Delta}$; AKP) and either transplanted directly into syngeneic mice, or re-cultured upon one round of orthotopic growth to promote further tumor progression in vivo (AKP$^{re}$). AKP$^{re}$ tumors indeed displayed more aggressive tumor growth with earlier onset and less differentiated morphology. However, deletion of $Zeb1$ in fibroblasts did not affect the engraftment of organoids, overall survival, primary tumor size, and tumor morphology upon transplantation of AKP (Figs. 1J and EV1A–D) or AKP$^{re}$ organoids (Fig. EV1E–K). Strikingly, spontaneous metastasis to the liver was decreased in Fib$^{\Delta Zeb1}$ mice, regardless of AKP or AKP$^{re}$ transplantation, as reflected in the fraction of metastasis-bearing mice and the number of metastases per mouse (Fig. 1K,L), indicating a pro-metastatic role of ZEB1 in CAFs of sporadic and progressed CRC. Overall, these findings suggest that ZEB1 in fibroblasts regulates colon cancer initiation and progression in a tumor context- and stage-dependent manner.

## Fibroblast diversity is reduced upon deletion of $Zeb1$

We applied scRNA-seq to gain insights into the cellular heterogeneity of CAFs and the transcriptional changes upon $Zeb1$ loss. AOM/DSS tumors were enzymatically dissociated, and CAFs were enriched by fluorescence-activated cell sorting (FACS) of CD326 (EPCAM)−, CD45−, and CD31− cells before sequencing (SORT-seq, Fig. 2A, left). Integrated clustering of cells from both genotypes was performed, followed by annotation using previously reported CAF signatures (Bartoschek et al, 2018; Elyada et al, 2019) (Fig. EV2A). Differential abundance analysis (Zhao et al, 2021) was performed for unbiased identification of genotype-specific differences. Prominent abundance differences were observed in two regions that were enriched for myCAF and iCAF-specific genes (Fig. 2B–D). Fib$^{Ctrl}$ cells were overrepresented in the myCAF region and Fib$^{\Delta Zeb1}$ cells in the iCAF region (Fig. EV2B–D). Genotypes were studied independently to avoid that Fib$^{\Delta Zeb1}$ cells are influenced by Fib$^{Ctrl}$ cells. Here, walktrap clustering resulted in eight and six CAF clusters in Fib$^{Ctrl}$ and Fib$^{\Delta Zeb1}$ tumors, respectively (Fig. 2E). Cross-correlation of transcriptomes indicated ambiguity in Fib$^{\Delta Zeb1}$ clusters number 1 and 4 (Fig. 2F). Furthermore, no $Zeb1$-deficient CAFs were matched to Fib$^{Ctrl}$ cluster 8, altogether suggesting impaired diversification and subtype specification in Fib$^{\Delta Zeb1}$ CAFs. The transcriptomes of Fib$^{Ctrl}$ CAFs correlated well with "iCAF" and "myCAF" archetypes described in pancreatic cancer (Elyada et al, 2019; Ohlund et al, 2017) (Fig. 2G). In

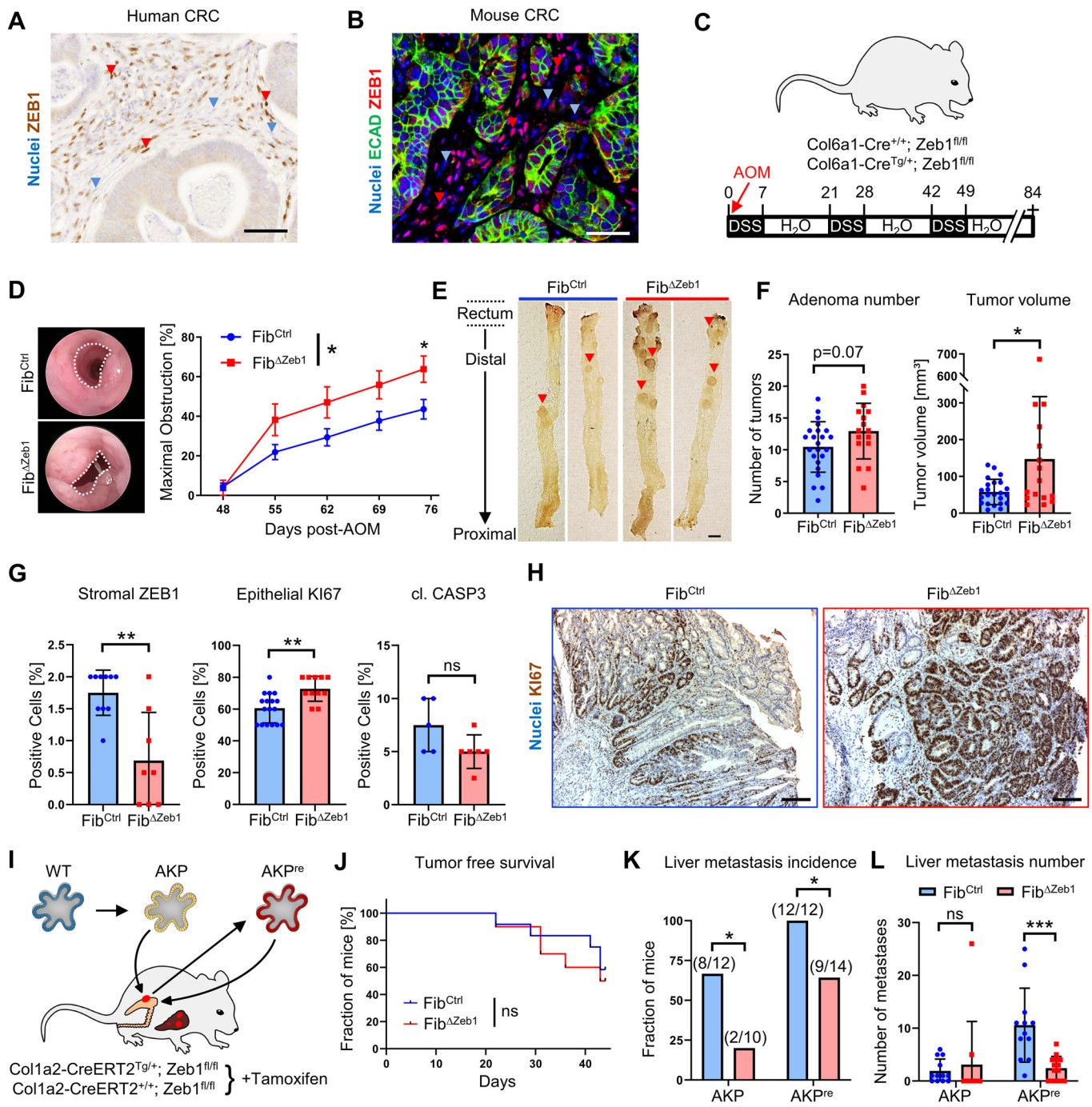

contrast, Fib^ΔZeb1 CAFs displayed less clusters with myCAF-like identity.

We also examined non-inflammation-driven orthotopic tumors by scRNA-seq. Following FACS, CAFs were analyzed in parallel to isolated CD45+ immune and EPCAM+ tumor cells (Fig. 2A, right). A separate analysis of genotypes resulted in two CAF clusters for Fib^Ctrl but only one for Fib^ΔZeb1 tumors (Fig. 2H). Individual analysis of CAFs confirmed that loss of *Zeb1* limits the CAF repertoire (Fig. EV2E) and cross-annotation confirmed increased ambiguity in Fib^ΔZeb1 (Fig. EV2F). We observed a higher

correlation of CAF clusters with "iCAF" and "myCAF" archetypes in Fib^Ctrl compared to Fib^ΔZeb1 (Fig. EV2G). Consistent with AOM/DSS CAFs, differential gene expression analysis between Fib^Ctrl and Fib^ΔZeb1 CAFs showed reduced signatures for ECM organization and increased inflammatory terms (Fig. 2I,J). In summary, our data show that independent of the CRC model, CAF diversification and specification is strongly impaired when ZEB1 is absent.

To study the differentiation defects in situ, we performed multiplex immunofluorescence staining. In orthotopic tumors, fibroblasts were defined as VIM+, CD45−, EPCAM− cells, and

**Figure 1. Context-dependent role of stromal ZEB1 during colorectal carcinogenesis.**

(A, B) ZEB1 IHC on human (A) and ZEB1/E-cadherin IF stainings on mouse (B) CRC samples. Red and blue arrowheads depict cells with high and low/absent ZEB1 detection, respectively. (C–H) AOM/DSS model (C) showing representative endoscopic images (D) (dotted lines indicate unobstructed area) and quantification of colon obstruction in Fib$^{Ctrl}$ and Fib$^{ΔZeb1}$ mice ($n = 24/17$ for Fib$^{Ctrl}$/Fib$^{ΔZeb1}$, Genotype: $P = 0.0282$, two-way ANOVA, day 75: $P = 0.0382$, Šídák's multiple comparisons test), macroscopic images (E, colons opened longitudinally, arrowheads point to individual tumors) and quantification of tumor volume and number (F) ($n = 23/16$ for Fib$^{Ctrl}$/Fib$^{ΔZeb1}$, number: $P = 0.0714$, volume: $P = 0.0180$, Student's $t$ test). Quantitative analysis of ZEB1 depletion, proliferation (KI67) and apoptosis (cl. Caspase 3) (G) with representative KI67 IHC is given (H). For ZEB1, the fraction of positive stromal cells was quantified. For KI67 and cl. Caspase (CASP) 3, the fraction of all tumor cells was quantified ($n = 17/11$ for Fib$^{Ctrl}$/Fib$^{ΔZeb1}$, Zeb1: $P = 0.0054$, KI67: $P = 0.0020$, cl. CASP3: $P = 0.1126$, Mann–Whitney test). (I–L) Schematic overview of orthotopic transplantation of tumor organoids into the cecum of Fib$^{Ctrl}$ and Fib$^{ΔZeb1}$ mice (I). AKP$^{re}$ organoids were generated after re-culturing cells retrieved from orthotopic AKP tumors. Analysis of tumor onset (detected by palpation) after orthotopic AKP organoid transplantation (J) ($n = 12/10$ for Fib$^{Ctrl}$/Fib$^{ΔZeb1}$, $P = 0.6401$, Mantel–Cox test) as well as of liver metastasis incidence (K) (AKP: $P = 0.0427$, AKP$^{re}$: $P = 0.0425$, Fisher's exact test) and numbers (L) after orthotopic transplantation of AKP or AKP$^{re}$ tumor organoids ($n = 12/10$ for Fib$^{Ctrl}$/Fib$^{ΔZeb1}$ with AKP and 12/14 for Fib$^{Ctrl}$/Fib$^{ΔZeb1}$ with AKP$^{re}$, AKP: $P = 0.8469$, AKP$^{re}$: $P = 0.0007$, two-way ANOVA). All mice with AKP$^{re}$ transplantation were treated with control IgG and are shown also in Fig. 5. Mice with AKP transplantation were treatment-naïve. Data information: Data are presented as mean ± SEM (D) or mean ± SD (F, G, L). Scale bars represent 50 μm (A, B), 5 mm (E) or 100 μm (H). Source data are available online for this figure.

CAFs were further characterized by αSMA, C3, and MHCII staining (Figs. 2K and EV3A,B). Quantitative image analysis and dimensionality reduction allowed classification into myCAF, iCAF, and apCAF-enriched populations (Fig. EV3C). While the majority of cells in Fib$^{Ctrl}$ displayed a myofibroblast-like phenotype, CAFs in Fib$^{ΔZeb1}$ tumors were predominated by the inflammatory subtype (Fig. 2L). Here, *Zeb1* loss did not affect apCAFs or result in a mixed αSMA+ /C3+ iCAF/myCAF identity (Fig. EV3A–D).

## ZEB1 is crucial for myofibroblast differentiation

We next studied the functional consequences of *Zeb1* loss and CAF polarization. We established adherent colon fibroblast cultures from *Zeb1*$^{fl/fl}$ mice, which showed a typical myofibroblast-like morphology. Induction of Cre recombinase activity resulted in efficient *Zeb1* deletion in vitro, which was confirmed by immunofluorescence staining (Fig. 3A). Of note, *Zeb1* deficiency-induced loss of fibrillary αSMA staining, demonstrating reduced myofibroblast differentiation and CAF activation (Sahai et al, 2020). Consistently, *Zeb1*-deleted fibroblasts showed lower expression of myofibroblast markers by qRT-PCR and bulk RNA sequencing analyses (Fig. 3B,C) and strongly reduced contraction in collagen (Fig. 3D). In contrast, we observed no major changes in proliferation or senescence of *Zeb1*-deficient fibroblasts (Appendix Fig. S3A,B). To study myofibroblast-specific functions in vivo, skin wound healing assays revealed delayed wound closure in Fib$^{ΔZeb1}$ compared to Fib$^{Ctrl}$ mice (Fig. 3E; Appendix Fig. S3C). Furthermore, the impaired myCAF polarization in AOM/DSS adenomas and orthotopic tumors was associated with reduced deposition of collagen (Fig. 3F,G). Since collagenous ECM can constitute a physical barrier for immune cell infiltration, we conducted an in vitro migration assay to test the ability of fibroblasts to block the passage of mouse splenocytes (Fig. 3H). Strikingly, Fib$^{ΔZeb1}$ fibroblasts showed a pronounced barrier defect. Collectively, these data demonstrate the fundamental role of ZEB1 in myofibroblast specification and function.

## Loss of *Zeb1* in fibroblasts induces a pro-inflammatory tumor microenvironment

To clarify whether the inflammatory polarization in combination with the reduced collagen deposition of *Zeb1*-deleted CAFs affects tumor immune cell infiltration, we performed IHC analyses.

Strikingly, in inflammation-induced AOM/DSS tumors, we found increased infiltration of CD4+ T cells and FOXP3+ Tregs, but not of CD8+ cells or F4/80+ macrophages into Fib$^{ΔZeb1}$ tumors (Fig. 4A), indicating modulation of adaptive anti-tumor immunity. Concomitantly, increased B-cell infiltration and an overall upregulation of the immune checkpoint molecule PD-L1 were observed, suggesting tolerance induction upon T-cell activation. Similar to AOM/DSS adenomas, increased T- and B-cell infiltration and expression of PD-L1 was observed in Fib$^{ΔZeb1}$ orthotopic tumors, reflecting sporadic CRC (Fig. 4B). Yet, in contrast to AOM/DSS tumors, CD8+ T cells and macrophages were enriched. Since the primary tumor size was comparable in Fib$^{Ctrl}$ and Fib$^{ΔZeb1}$ orthotopic tumors (Fig. EV1C,H,K), the observed T-cell infiltration apparently did not result in an effective anti-tumor response, presumably due to T-cell inactivity.

To investigate if the differences between both models can be explained by the inflammatory environment in the AOM/DSS model, we employed another model of sporadic CRC, in which AOM-induced mutagenesis in combination with epithelial cell-specific *Tp53* deletion results in invasive CRC (Neufert et al, 2021; Schwitalla et al, 2013) (Fig. EV4A). Here, deletion of *Zeb1* in fibroblasts resulted in fewer, smaller and less invasive tumors, indicating delayed tumor progression in Fib$^{ΔZeb1}$ mice in the AOM/ p53 model (Fig. EV4B–D). We observed enhanced infiltration of CD4+ and CD8+ cells but no and only trending increase in FOXP3+ Tregs and stromal PD-L1 levels, respectively (Fig. EV4E). Collectively, our data show a context-dependent immunomodulatory role of ZEB1 expression in fibroblasts. This notion was supported by analysis of AOM/DSS tumors at day 50 during the acute phase of inflammation. Even before overt tumor outgrowth, significantly increased CD8+ and FOXP3+ cell infiltration was observed (Fig. EV4F–H), suggesting that the acute inflammation triggers a compensatory regulation of the immune microenvironment upon *Zeb1* loss in fibroblasts.

To explore the mechanism for the altered immune cell infiltration in *Zeb1*-deficient CAFs, we tested the response of cultured fibroblasts to the chemokine IL1α, a crucial stimulator of inflammatory polarization in CRC CAFs (Nicolas et al, 2022). We monitored basal and IL1α-induced NFκB activation in cultured Fib$^{Ctrl}$ and Fib$^{ΔZeb1}$ fibroblasts by qRT-PCR and Western blot analyses. Upon stimulation, Fib$^{ΔZeb1}$ fibroblasts showed increased expression of NFκB targets including *Ccl2* and *Cxcl1* that both have been described as iCAF markers (Biffi et al, 2019; Elyada et al, 2019;

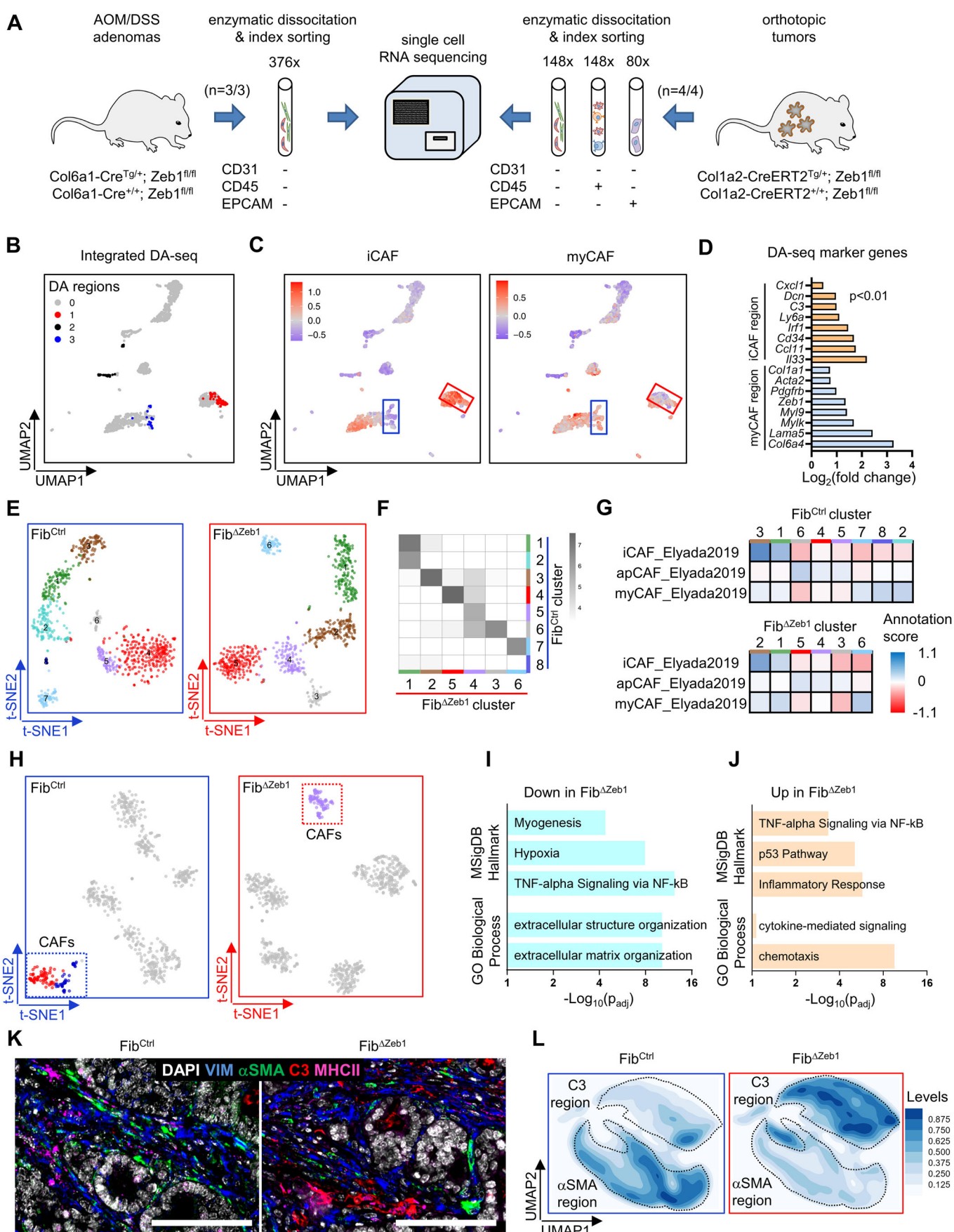

Figure 2.   scRNA sequencing of CAFs reveals a key role of ZEB1 for fibroblast plasticity.

(A) Experimental scheme for isolation of CAFs and other cell types. AOM/DSS adenomas (left) and primary tumors from the orthotopic AKP model (right) of Fib^Ctrl and Fib^ΔZeb1 mice were enzymatically dissociated, and fibroblasts were enriched by depletion of CD31+, CD45+, and EPCAM+ cells by flow cytometry. Numbers of analyzed mice and sorted cells are shown. (B–G) Fibroblast analysis from the AOM/DSS model. (B) Integrated UMAP-(Leiden) clusters of CAFs from n = 3 mice per genotype were subjected to differential abundance ('DA') analysis. 3 significant DA regions between Fib^Ctrl and Fib^ΔZeb1 were found (Wilcoxon P values for regions 1, 2, and 3: 5.37 × 10^−14; 1.35 × 10^−12 and 5.62 × 10^−06). Note that region 2 derived exclusively from one mouse and was therefore neglected. (C) Projections of iCAF and myCAF gene signature scores (Elyada et al, 2019). Note the highlighted DA regions from (B). (D) Log_2 fold changes of representative DAseq marker genes in DA regions 1 and 3, designated as "iCAF" and "myCAF" regions, respectively (P < 0.01, as determined by STG (stochastic gates) within DAseq). (E) Independent t-SNE clustering in Fib^Ctrl (left) and Fib^ΔZeb1 mice (right). (F) Cluster similarities defined by cluster annotations based on "SingleR" scores (see "Methods" for details) (Aran et al, 2019). Grayscale shows the log_2-transformed numbers of Fib^ΔZeb1 cells assigned to the different Fib^Ctrl clusters. (G) Heatmap showing the similarity (annotation scores) of gene expression in CAF clusters with published gene sets. (H–J) scRNA-seq of fibroblasts, immune cells and tumor cells after orthotopic transplantation of AKP organoids. (H) Data from n = 4 mice per genotype of the orthotopic model were subjected to scRNA sequencing and t-SNE clustering in Fib^Ctrl (left) or Fib^ΔZeb1 tumors (right). (I, J) Down- and upregulated gene sets in Fib^ΔZeb1 CAFs in comparison to the two Fib^Ctrl CAF clusters, as determined using Enrichr. Differential genes between Fib^ΔZeb1 CAFs and each of the two Fib^Ctrl CAF clusters were individually determined and pooled before enrichment analysis (FDR ≤ 0.1; P ≤ 0.05, as determined by default Welch t tests within the findMarkers function). (K, L) Tumor sections from mice after transplantation with AKP organoids were subjected to multiplexed immunostaining (n = 5/6 for Fib^Ctrl/Fib^ΔZeb1). (K) Representative images of fibroblast and CAF markers (VIM, αSMA, C3, MHCII). (L) Individual cells were UMAP embedded based on their CAF marker intensities and the density distribution of cells from Fib^Ctrl or Fib^ΔZeb1 mice is shown. Data information: Scale bars represent 100 μm (K). Source data are available online for this figure.

Nicolas et al, 2022) (Fig. 4C). Consistently, strongly increased levels of phosphorylated NFκB and IκBα were observed in lysates of Fib^ΔZeb1 fibroblasts that were stimulated with IL1α for 15 min (Fig. 4D), when Fib^Ctrl cells already partially resolved the signal, suggesting an overshooting pathway activation. Indeed, immunostaining of phospho-NFκB showed increased pathway activity in both Fib^ΔZeb1 AOM/DSS and orthotopic tumors (Fig. 4E). Furthermore, secretome analysis in cultured CAFs from AOM/DSS tumors identified increased basal levels of CCL2 upon Zeb1 loss (Appendix Fig. S4A,B), which was confirmed by IHC (Appendix Fig. S4C). Notably, increased phospho-NFκB and CCL2 staining were also observed during the acute inflammation in Fib^ΔZeb1 mice at day 50 of AOM/DSS treatment (Appendix Fig. S4D,E). Our data suggest that elevated cytokine production from Zeb1-deficient CAFs may act as a potent lymphocyte chemoattractant. Thus, the influence of Fib^Ctrl or Fib^ΔZeb1 fibroblasts on T-cell recruitment was tested in a transwell migration assay using stimulated T cells in co-culture with fibroblasts, where migration was significantly increased towards a Fib^ΔZeb1 fibroblast monolayer (Fig. 4F).

As a plastic cell population, the observed changes could result from an intrinsic differentiation bias, or a failure to respond to external signals. To tackle this question, we seeded CAFs in a 3D ECM, representing a quiescent status, or applied IL1α or TGFβ treatment, to direct them into the myCAF and iCAF lineages, respectively (Ohlund et al, 2017). qRT-PCR showed that under baseline conditions Fib^ΔZeb1 expressed unaffected levels of iCAF markers (Ccl2 and Cxcl1) but significantly reduced levels of myCAF markers (Acta2 and Tagln) (Fig. 4G). While the induction of inflammatory genes by IL1α was comparable in this setting (Fig. 4H), after TGFβ addition Fib^ΔZeb1 CAFs showed a striking defect to downregulate inflammatory genes (Fig. 4I). In contrast, the TGFβ-mediated induction of myCAF markers was less prominently affected. Collectively, our data indicate that ZEB1 limits the sensitivity of fibroblasts to NFκB-activating signals, which causes an iCAF bias affecting tumor immune infiltration.

## Loss of Zeb1 in CAFs enhances the response of CRC to immune checkpoint blockade

Given the refractoriness of non-hypermutated CRC to ICB therapy, the increased immune cell infiltration and induction of checkpoint

molecules observed in Fib^ΔZeb1 mice may point to a strategy to render MSS tumors more sensitive. We investigated this hypothesis in orthotopic and AOM/DSS tumors. Following transplantation of AKP^pre organoids, tumor-bearing mice were injected subcutaneously with anti-PD-L1 antibodies for 3 weeks. Strikingly, anti-PD-L1 therapy but not control IgG significantly delayed tumor growth of Fib^ΔZeb1 in comparison with Fib^Ctrl mice, eventually resulting in smaller Fib^ΔZeb1 tumors at the endpoint (Fig. 5A,B). In addition, infiltration of CD4+, CD8+ T cells, and B cells was further increased by ICB in tumors from treated Fib^ΔZeb1 mice, whereas FOXP3+ cells were not affected. PD-L1 expression was reduced in Fib^ΔZeb1 mice upon ICB, indicating successful reactivation of the adaptive immune response (Fig. EV5A). In a more clinically relevant setting, we explored whether ICB sensitivity is also induced by later Zeb1 deletion, when tumors have already formed. For this purpose, we inactivated Zeb1 in orthotopic tumors simultaneously with ICB administration (Fig. EV5B–D). Although not significant, a similar trend towards smaller tumors was observed in Fib^ΔZeb1 mice, when Zeb1 was depleted from a preformatted immune TME. In autochthonous AOM/DSS tumors, we employed dual ICB targeting PD-L1 and CTLA-4, as we found a substantial increase in the number of Tregs in tumors of Fib^ΔZeb1 mice in this model (Fig. 4A). Tumor growth was monitored longitudinally via endoscopy, and adenoma-bearing mice were subjected to ICB after recovery from cyclic DSS-induced inflammation at day 70. Strikingly, dual ICB abrogated tumor growth in Fib^ΔZeb1 mice combined with a more robust increase in CD8+ T cells, ablation of FOXP3+ cells, and reduced PD-L1 expression (Figs. 5C,D and EV5E). Together, our analyses demonstrate that loss of Zeb1 in fibroblasts induces sensitivity to ICB in both colitis-induced and sporadic tumor models.

## Discussion

Functional diversification of CAFs is a result of coevolution with tumor cells in the TME. Using autochthonous and organoid transplantation models of CRC, we have discovered a key role of ZEB1 in governing this plasticity. Zeb1 deletion severely impairs myofibroblastic and modulates inflammatory CAF functions, jointly affecting immune cell infiltration. Consequently, inflammation-driven adenoma formation is augmented, yet more progressed cancer models show reduced invasiveness and

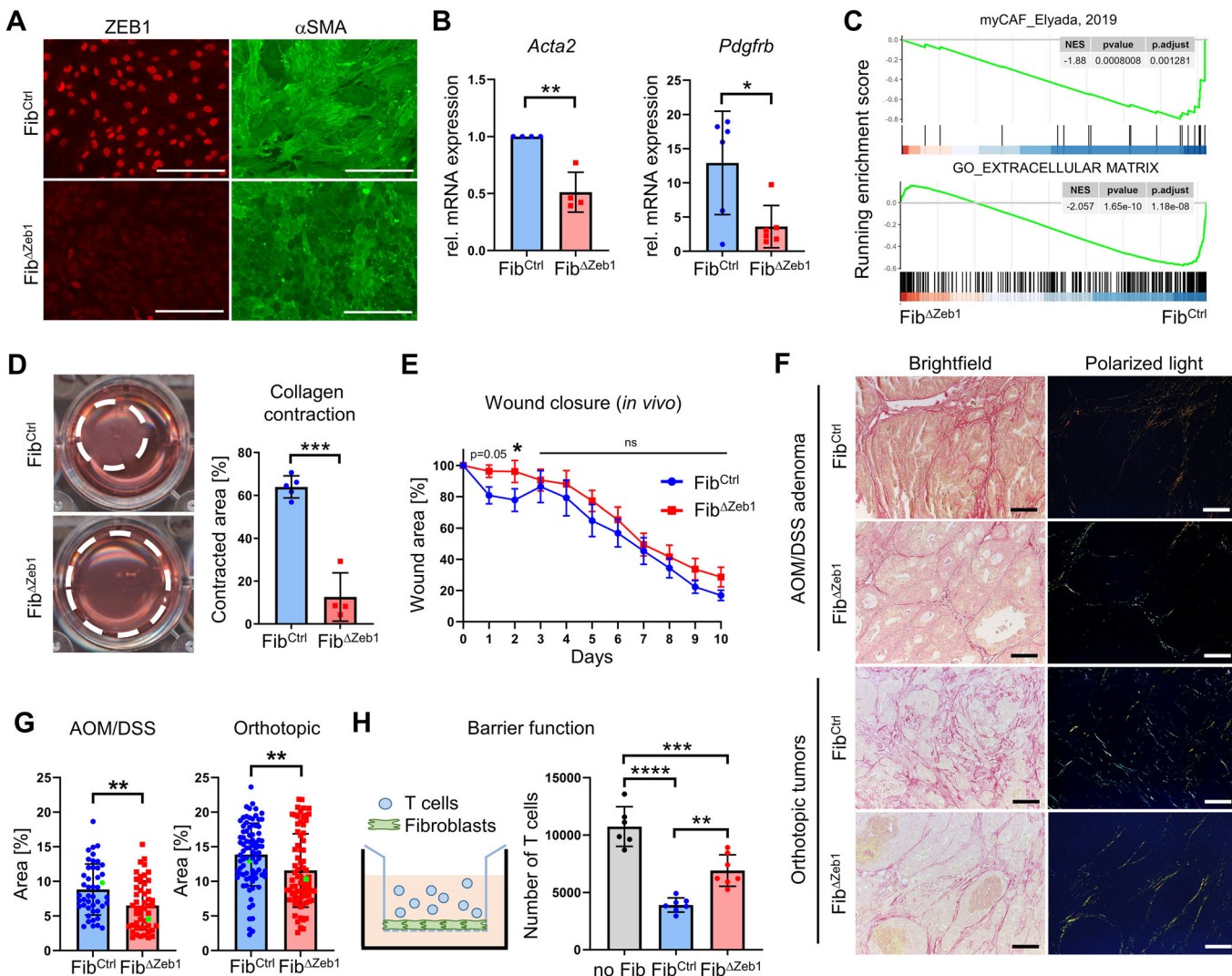

**Figure 3. ZEB1 is critically involved in myofibroblast differentiation and functionality.**

(A) Representative IF staining of ZEB1 and αSMA in fibroblasts after in vitro recombination. (B) qRT-PCR analysis of myofibroblast markers *Acta2* and *Pdgfrb* in fibroblasts after in vitro recombination, ($n = 4/6$; 2 independent fibroblast lines per genotype in 2/3 biological replicates for *Acta2/Pdgfrb, Acta2*: $P = 0.0014$, *Pdgfrb*: $P = 0.0190$, Student's $t$ test). (C) Gene set enrichment analysis using myCAF and extracellular matrix (GO: 0031012) signatures. (D) Representative images and quantification of collagen contraction assay ($n = 5/4$ independent Fib$^{Ctrl}$/Fib$^{ΔZeb1}$ lines; $P < 0.0001$, Student's $t$ test). (E) Quantification of relative wound area in Fib$^{Ctrl}$ and Fib$^{ΔZeb1}$ mice during a skin wound healing model ($n = 16/22$), day 1: $P = 0.0539$, day 2: $P = 0.0326$, two-way ANOVA). (F, G) Representative images (F) and quantification (G) of Picrosirius red staining of tumor sections from Fib$^{Ctrl}$ and Fib$^{ΔZeb1}$ mice after AOM/DSS and orthotopic transplantation (45/50 image sets derived from $n = 13/9$ Fib$^{Ctrl}$/ Fib$^{ΔZeb1}$ mice for AOM/DSS and 84/76 image sets derived from $n = 8/9$ Fib$^{Ctrl}$/Fib$^{ΔZeb1}$ mice for the orthotopic tumor model, AOM/DSS: $P = 0.0023$, orthotopic: $P = 0.0015$, Mann–Whitney test). Areas from representative images are marked in green. (H) Quantification of T-cell migration through a transwell insert alone or with a layer of fibroblasts after in vitro recombination of *Zeb1* ($n = 7$ independent lines, no Fib vs Fib$^{Ctrl}$: $P < 0.0001$, no Fib vs Fib$^{ΔZeb1}$: $P = 0.0002$, Fib$^{Ctrl}$ vs Fib$^{ΔZeb1}$: $P = 0.0012$, Tukey's multiple comparisons test). Data information: Data are presented as mean ± SD (B, D, G, H) or mean ± SEM (E). Scale bars represent 200 μm (A) or 80 μm (F). Source data are available online for this figure.

metastasis. Despite this stage-dependent outcome, the common immunomodulation leads to checkpoint activation in Fib$^{ΔZeb1}$ tumors. We show that targeting ZEB1 sensitizes to ICB reinforcing a translational rationale to target specific CAF subtypes for therapy of MSS tumors.

Our data identify that ZEB1 in CAFs acts as an immunosuppressor that promotes malignant progression. As both archetypes, i.e., iCAFs and foremost myCAFs, are strongly dysregulated in the absence of ZEB1, we conclude that ZEB1 controls general CAF plasticity. This is supported by our single-cell analysis in both models,

the impaired myofibroblastic functionality and the augmented response to inflammatory stimuli in vitro. While previous reports have established that EMT(-related-) TFs ZEB1, SNAIL, TWIST, and PRRX1 can regulate the classical mechanoinvasive features of CAFs directly affecting tumor malignancy (Feldmann et al, 2021; Stanisavljevic et al, 2015; Yeo et al, 2018), our data highlight a critical impact of CAF plasticity on modulation of anti-tumor immunity that may contribute to the poor prognosis reported in breast and pancreatic cancer patients with elevated ZEB1+ stromal cells (Bronsert et al, 2014; Ouled Dhaou et al, 2020).

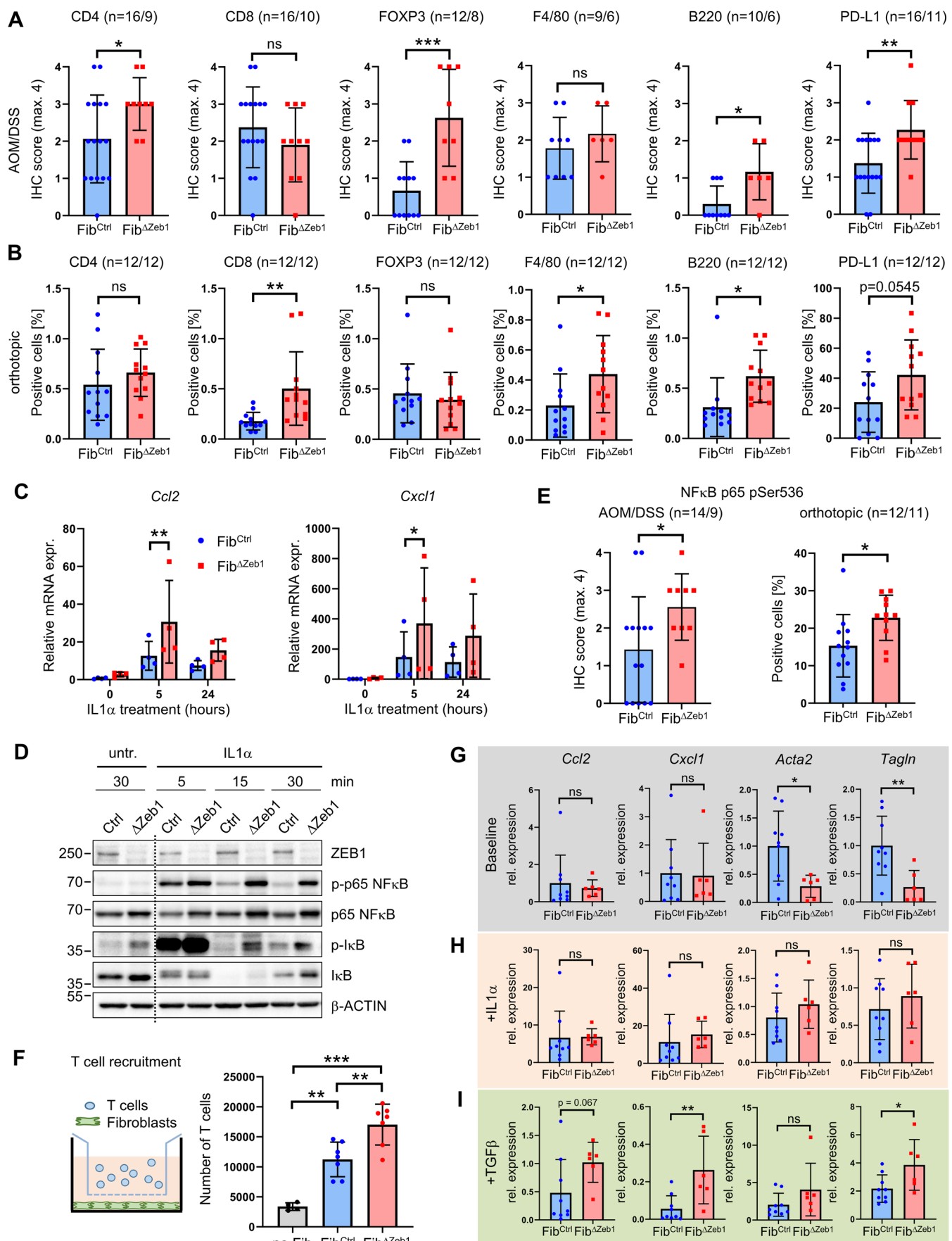

◄ **Figure 4. ZEB1 attenuates inflammatory signaling in fibroblasts and limits immune cell infiltration in multiple CRC models.**

(A, B) IHC-based quantification of immune cell infiltration and PD-L1 expression in tumors from Fib[Ctrl] and Fib[ΔZeb1] mice in the AOM/DSS model (A) and after orthotopic transplantation of AKP tumor organoids (B). Number of experimental mice per genotype are indicated (CD4, CD8, FOXP3, F4/80, B220, PD-L1: $P = 0.0414, 0.2745, 0.0005, 0.3751, 0.0134, 0.0082$ (AOM/DSS), $P = 0.3383, 0.0068, 0.5852, 0.0401, 0.0122, 0.0545$ (orthotopic), Student's $t$ test). (C) qRT-PCR analysis of *Cxcl1* and *Ccl2* mRNA expression in fibroblasts after in vitro recombination of *Zeb1* and stimulation with IL1α ($n = 4$, two independent fibroblast lines per genotype in two biological replicates, 0, 5, 24 h: $P = 0.9248, 0.0070, 0.1816$ (*Ccl2*), $P = 0.9998, 0.0337, 0.0890$ (*Cxcl1*), Šídák's multiple comparisons test). (D) Western blot of NFκB pathway activity in fibroblasts after 15 min of IL1α stimulation. β-ACTIN detection was used as a loading control. (E) IHC-based quantification of phospho-NFκB p65 (Ser536) in mice after AOM/DSS tumorigenesis or transplantation of AKP[re] organoids. Numbers of experimental mice per genotype are indicated (AOM/DSS: $P = 0.0435$, orthotopic: $P = 0.0238$, Student's $t$ test). (F) Quantification of T-cell attraction to fibroblasts after in vitro recombination of *Zeb1* ($n = 7$ independent lines, no Fib vs Fib[Ctrl]: $P = 0.0014$, no Fib vs Fib[ΔZeb1]: $P < 0.0001$, Fib[Ctrl] vs Fib[ΔZeb1]: $P = 0.0043$, Tukey's multiple comparisons test). (G-I) Fib[Ctrl] and Fib[ΔZeb1] fibroblasts were seeded in 3D ECM and expression of fibroblast subtype markers was quantified by qRT-PCR. Baseline marker expression of Fib[Ctrl] and Fib[ΔZeb1] fibroblasts without additional treatment (G) and after treatment with IL1α (H) or TGFβ (I) relative to the respective untreated condition ($n = 9/6$ for Fib[Ctrl]/Fib[ΔZeb1] fibroblast lines, *Ccl2, Cxcl1, Acta2, Tagln*: $P = 0.6830, 0.8861, 0.0184, 0.0084$ (baseline), $P = 0.9318, 0.5463, 0.3182, 0.7161$ (+ IL1α), $P = 0.0670, 0.0077, 0.1469, 0.0346$ (+ TGFβ), Student's $t$ test). Data information: Data are presented as mean ± SD (A-C, E-I). Source data are available online for this figure.

As mechanistic basis for enhanced inflammation and immune cell infiltration, we show reduced collagen deposition, increased inflammatory signaling and chemoattraction in *Zeb1*-deficient CAFs collectively modulating the immune TME. The stage-dependent consequences can be explained by the known tumor-promoting role of inflammation in the AOM/DSS model (Greten et al, 2004; Koliaraki et al, 2015; Neufert et al, 2007; Neufert et al, 2021; Schmitt and Greten, 2021). Consistently, phospho-NFκB and CCL2 levels were upregulated already during the acute phase of inflammation, and we found strongly increased infiltration of B cells, CD4+ T cells and FOXP3+ Tregs in Fib[ΔZeb1] tumors accompanied by enhanced PD-L1. CD8+ T cells were unaffected, yet strongly induced upon dual ICB, indicating an immune checkpoint involving Tregs. This agrees with previous reports on tolerance induction in AOM/DSS and colitis-associated CRC (Olguin et al, 2018; Pastille et al, 2014; Yassin et al, 2019). In contrast, increased CD8+ T cells and delayed tumor progression in the sporadic Fib[ΔZeb1] tumors showed an unaffected FOXP3+ infiltration, pointing towards Tregs as part of a compensatory immunomodulatory mechanism in the AOM/DSS model. Taken together, we conclude that loss of *Zeb1* in CAFs facilitates inflammation, immune infiltration, and co-activated immunosuppression.

We furthermore identified ZEB1 as an important regulator balancing myofibroblastic and inflammatory functions of CAFs. Of note, TGFβ signaling has been shown essential for the acquisition of classical myofibroblast phenotypes (Biffi et al, 2019; Elyada et al, 2019; Sahai et al, 2020; Tauriello et al, 2018; Tauriello et al, 2022) and in tumor cells, ZEB1 is a key mediator of TGFβ signaling (Krebs et al, 2017; Schuhwerk et al, 2022; Stemmler et al, 2019). Our data indicates that ZEB1 acts downstream of TGFβ signaling and facilitates myofibroblast polarization at least in part by downregulation of inflammatory gene expression. Consistently, *Zeb1*-deleted fibroblasts displayed increased NFκB activation in vitro and in vivo. In this regard, it is important to mention that ZEB1 has been shown as a direct transcriptional (co-)inducer of inflammatory gene expression, like IL6, IL8 and others in several cell types, such as breast cancer cells and fibroblasts (Fu et al, 2019; Katsura et al, 2017), corneal fibroblasts, hematopoietic and myeloid cells (Cortes et al, 2017; Liang et al, 2022; Qian et al, 2021; Scott and Omilusik, 2019; Wang et al, 2009). Together these data suggest that ZEB1 elicits context-dependent effects on inflammation.

The sensitization of unresponsive tumors to ICB that was observed in two independent mouse CRC models points towards a more general CAF-engaging immune checkpoint in CRC, in line with the refractoriness of non-hypermutated tumors (Le et al, 2015; Tauriello et al, 2022). Thus, ZEB1 expression in CAFs may serve as a negative predictive marker for ICB efficacy and agents that interfere with ZEB1 function may be beneficial to enhance immune infiltration and ICB sensitivity. Yet, before clinical translation is possible several limitations need to be considered: Because pharmacologic targeting of ZEB1 as a TF is challenging, new strategies will be required such as the development of PROTACs. Alternatively, interference with ZEB1 downstream programs could allow to modulate CAF identities. For instance, inhibiting the DNA damage response (DDR) kinase ATM has recently been shown to inhibit myofibroblastic features, increase immune infiltration and sensitize to immune checkpoint therapy in subcutaneous tumor models (Mellone et al, 2022). In this regard, we recently discovered an actionable vulnerability in ZEB1[high] cancer cell sub-populations by inhibiting the DDR nuclease MRE11 (Schuhwerk et al, 2022). In addition, a more detailed understanding will be essential, how CAF polarization affects the tumor immune environment in patients. Here in particular, the impact of standard therapies should be considered, because radiotherapy-induced senescence in iCAFs favors therapy resistance and a poor outcome in rectal cancer (Nicolas et al, 2022). Likewise, ablation of αSMA-high myCAFs, or deletion of type 1 collagen in myofibroblasts was shown to aggravate disease course in PDAC and experimental liver metastasis (Bhattacharjee et al, 2021; Chen et al, 2021; Ozdemir et al, 2014). Hence, targeting regulators of CAF plasticity, rather than fully depleting integral components of the ECM or CAFs in general, may be beneficial. Our study suggests that reduction of ZEB1 in CAFs may turn immunologically "cold" into "hot" CRCs and thereby sensitize patients to ICB. Given the known role of ZEB1 in tumor cells to induce EMT, stemness, and chemoresistance, combined targeting of ZEB1 in fibroblast and tumor cells might act synergistically to improve CRC therapy. However, correct timing may be crucial to allow reformatting of the immune TME for efficient sensitization to ICB.

# Methods

## Ethics statement

Animal husbandry and all experiments were performed according to the European Animal Welfare laws and guidelines. The protocols

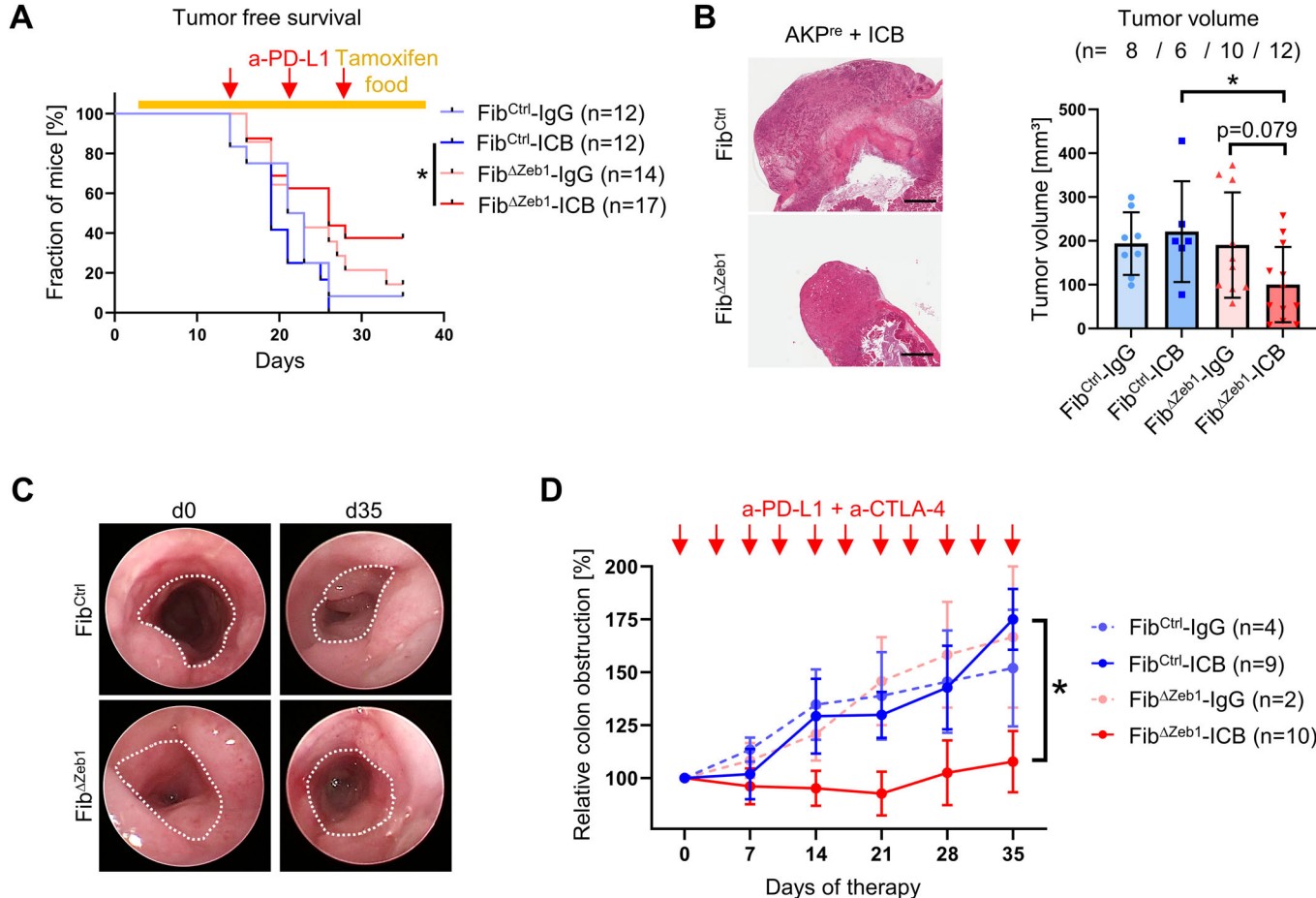

**Figure 5. Loss of *Zeb1* in fibroblasts enables response to ICB therapy.**

(A) Kaplan–Meier analysis showing tumor-free survival of Fib$^{Ctrl}$ and Fib$^{\Delta Zeb1}$ mice after orthotopic transplantation of AKP$^{re}$ organoids and intraperitoneal injection of anti-(a)-PD-L1 antibodies or control IgGs, as indicated by red arrows. Recombination of *Zeb1* was induced by tamoxifen food starting from day (d) 3. Mice were considered tumor-free if no tumor was detected by palpation. Numbers of experimental mice per condition are indicated (Fib$^{Ctrl}$-ICB vs Fib$^{\Delta Zeb1}$-ICB: $P = 0.0189$, Mantel–Cox test). (B) Representative H&E images and quantification of tumor volumes after orthotopic transplantation of AKP$^{re}$ tumor organoids and ICB. Only tumors collected after d28 were included in this analysis. Number of experimental mice per condition are indicated. IgG-treated mice contributed to the initial AKP$^{re}$ analysis (Fib$^{Ctrl}$-ICB vs Fib$^{\Delta Zeb1}$-ICB: $P = 0.0397$, Fib$^{\Delta Zeb1}$-IgG vs Fib$^{\Delta Zeb1}$-ICB: $P = 0.0792$, Šídák's multiple comparisons test). (C, D) ICB in the AOM/DSS model, applied by intraperitoneal injection of a-PD-L1 and a-CTLA-4 antibodies or control IgGs. Starting at d70 of AOM/DSS tumorigenesis antibodies were administered two times/week to all mice with at least 25–30% colon obstruction. This time point corresponds to d0 of ICB. Representative endoscopic images at d0 and d35 of ICB in the AOM/DSS model. Dotted line indicates unobstructed areas (C). Quantification of colon obstruction relative to d0 (D). Numbers of experimental mice per condition are indicated (Fib$^{Ctrl}$-ICB vs Fib$^{\Delta Zeb1}$-ICB: two-way ANOVA, day 35: $P = 0.0211$. Data information: Data are presented as mean ± SD (B) or mean ± SEM (D). Scale bars represent 1 mm (B). Source data are available online for this figure.

were approved by the committee on ethics of animal experiments of Bavaria (Regierung von Unterfranken, Würzburg; TS-18/14, 55.2-DMS-2532-2-270, -2-952, and -2-1133) and of Hessen, Germany (Regierungspräsidium Darmstadt F123/1031, F123/1040, and F123/2001). Power analysis was used to calculate the sample size required for animal experiments. Animals were kept on a 12:12 h light-dark cycle and provided with food and water ad libitum in the animal facilities of the Friedrich-Alexander University of Erlangen-Nürnberg and the Georg-Speyer-Haus Frankfurt. Col1a2-CreERT2$^{Tg/+}$;*Zeb1*$^{fl/fl}$ and Col6a1-Cre$^{Tg/+}$;*Zeb1*$^{fl/fl}$ mice were kept on FvB and C57BL/6 backgrounds, respectively. Col6a1-Cre$^{Tg/+}$;*Zeb1*$^{fl/fl}$;Vil-Flp$^{Tg/+}$;*Tp53*$^{FRT/FRT}$ Fib$^{\Delta Zeb1}$ mice were kept on a mixed FvB/C57BL/6 background.

## Mouse models of CRC

Rosa26-tdTomato (Madisen et al, 2010); RRID:IMSR_JAX:007914) and Rosa26-mTmG (Muzumdar et al, 2007); RRID:IMSR_JAX:007576) mice have been described previously. Conditional *Zeb1* knockout mice (Brabletz et al, 2017); MGI:5901939) were crossed with mice expressing Col6a1-Cre (Armaka et al, 2008); GI:3775430) or Col1a2-CreERT2 (Zheng et al, 2002); RRID:IMSR_JAX:029567) to generate Cre-positive Fib$^{\Delta Zeb1}$ mice or Cre-negative Fib$^{Ctrl}$ mice. Age-matched littermates of both sexes were used for experiments and randomly assigned to groups within genotypes. Genotyping primer sequences are listed in Appendix Table S1.

AOM/DSS driven adenoma were induced as described (Neufert et al, 2007). Briefly, mice were injected intraperitoneally (i.p.) with 10 mg/kg of azoxymethane (AOM, Sigma, A5486) in 0.9% NaCl), prior to administration of three cycles of 1.75% dextran sulfate sodium (DSS, MP Biomedicals, SKU-0216011080) in the drinking water ad libitum, each interrupted by 2 weeks of regular water and finally sacrificed at day 84 or at ethical endpoints. A separate cohort of mice was sacrificed at day 50 at the peak of inflammation. Colonic inflammation and tumor growth was monitored by endoscopy under isoflurane anesthesia employing the Colorview endoscopic system (Karl Storz), as described before (Becker et al, 2006), expressed as numbers of adenoma and/or the maximal colon obstruction by the adenoma in percent of the full colonic diameter at the most blocked site. Colonic inflammation was visually scored from '0' (zero) to '4' (four), with '0' representing normal, '1' signs of thickening, increased vascularity, and/or fibrin visible, moderate granularity and/or stool being still shaped, '2' stronger thickening, vascularization, granularity and unshaped stool (no blood), '3' loss of transparency, extreme granularity, rectal blood (clots) and/or traces of blood in the mucosa and/or in the unshaped or spread stool, and '4' (partially) liquid stool, several sites of bloody mucosal damage or non-rectal liquid blood in the colon. Stool consistency was scored individually from '1' to '4' as well, with '0' being normal and solid, '1' softer than usual, '2' unshaped, '3' almost liquid but containing solid pieces of stool and/or containing traces of (clotted) blood, '4' being liquid and/or mostly bloody. For immune checkpoint inhibition, mice were injected i.p. with anti-PD-L1 (BXC-BE0101) and anti-CTLA-4 (BXC-BE0131) antibodies or isotype controls (BXC-BE0094 and BXC-BP0087) twice per week (10 mg/kg each, Bio X cell). Therapies started at day 70 for all mice, unless colon obstruction by adenomas did not reach 25–30%, as determined by endoscopy. Adenoma growth was then monitored weekly via endoscopy, and mice were sacrificed at day 105.

The AOM/p53 model was based on the previously reported Tp53ΔIEC model (Neufert et al, 2021; Schwitalla et al, 2013), using Villin-Flp mediated recombination and intestinal epithelium-specific inactivation of biallelic FRT-flanked Tp53 (intron 1 and 10). Experimental animals were generated by sequential crossings to generate experimental Col6a1-Cre$^{Tg/+}$;Zeb1$^{fl/fl}$;Vil-Flp$^{Tg/+}$;Tp53$^{FRT/FRT}$ Fib$^{ΔZeb1}$;Δp53 and Col6a1-Cre$^{+/+}$;Zeb1$^{fl/fl}$;Vil-Flp$^{Tg/+}$;Tp53$^{FRT/FRT}$ Fib$^{Ctrl}$;Δp53 littermates. AOM/p53 tumors were induced by six weekly doses of AOM (10 mg/kg, i.p.) in 0.9% NaCl, as described previously (Neufert et al, 2021; Schwitalla et al, 2013). Tumor growth was monitored by endoscopy as in the AOM/DSS model and animals were sacrificed after 161 days or at any other ethical endpoint.

Orthotopic transplantation of tumor organoids was performed as described (Fumagalli et al, 2018). In brief, AKP or AKP$^{re}$ organoids were mechanically dissociated, seeded in high-concentration collagen pads (6.5 mg/mL) and allowed to recover for 48 h. Collagen pads were transplanted below the *muscularis externa* of the mouse cecum, and serosal wounds were covered by an anti-adhesion barrier. Recombination of Zeb1 was induced at day 14 for AKP organoids and at day 4 for AKP$^{re}$ organoids by 400 mg/kg tamoxifen diet (custom order based on Altromin 1824P). Tumor growth was monitored three times a week by palpation, and mice were sacrificed when tumors exceeded 1 cm in diameter. For immune checkpoint inhibition, mice were injected i.p. with anti-PD-L1 (Bio X cell, BE0101) or matched control antibodies (Bio X cell, BE0090) at days 14, 21, and 28. To ensure

equal treatment of mice, only tumors collected after day 28 were considered for size comparison. For AKP$^{re}$ transplantation, only a small batch of untreated mice was used, and most analyses were performed in mice treated with control antibodies as mentioned in the figure legends for the reduction of mouse numbers.

## Skin wound healing

Wound closure of a skin excision wound in mice was determined for 10 days (Lin et al, 2011). Briefly, circular wounds were applied under sterile conditions in anaesthetized mice into the dorsum after depilation. The dorsal skin was lifted at the midline and punched through two layers of skin using a disposable biopsy punch (6 mm in diameter, Kai medical, Solingen #BP-60F). Wound size was determined at indicated time points with a digital calliper and by transferring wound outline to a transparent film for area calculation using ImageJ.

## Organoid culture and genetic engineering

Primary mouse intestinal organoid cultures were established as reported (Fan et al, 2019). Briefly, mouse colons were cut into small fragments, washed with PBS, and epithelial crypts were dissociated by incubation with 10 mM EDTA, passed through a 100 μm mesh and collected by centrifugation. Colon crypts were seeded at high density in BME (R&D Systems, 3533-010-02) and cultured in Advanced DMEM/F12 (Thermo Fisher Scientific, 12634028) with 20% Wnt3a conditioned medium, 10% Noggin conditioned medium, 10% R-spondin1 conditioned medium, B27 supplement (Thermo Fisher Scientific, 17504-044), 500 mM N-Acetylcysteine (Merck, A9165), 500 μg/mL human EGF (Peprotech, AF-100-15) and 500 μM A83-01 (Tocris, 2939/10). Established organoids were passaged twice a week by mechanical dissociation and reseeding.

For generation of tumor organoids, colons from Lox-Stop-Lox-Kras$^{G12D}$ mice were used (Jackson et al, 2001). Both Apc and Tp53 alleles were mutated by co-transfection of Cas9 with the respective sgRNA plasmids. The oncogenic Kras$^{LSL.G12D}$ allele was recombined by pAC1-Cre (ATCC, 39532) transfection. Modified lines were clonally expanded, and successful modification confirmed by Sanger sequencing. The hCas9 and gRNA_GFP-T2 plasmids were a kind gift from George Church (Addgene plasmids #30205 and #41820) and the Apc sgRNA plasmid kindly provided by Hans Clevers (Schwank et al, 2013). sgRNA oligonucleotides targeting Tp53 (sense: 5'-GTTTTAGAGCTAGAAATAGCAAG-3' antisense: 5'-AGTGAAGCCCTCCGAGTGTCGGTGTTTCGTCCTTTCCA-CAAGAT-3') were inserted into gRNA_GFP-T2 plasmid by inverse cloning as published (Schwank et al, 2013).

Tumor organoids from established tumors were obtained by enzymatic digestion of primary tumors with 0.1% Collagenase I (Thermo Fisher Scientific, 17018029), 0.2% Dispase II (Merck, D4693) and 2 U/mL DNase I (New England Biolabs, M0303L). Cells were passed through a 40 μm mesh and seeded at high density in BME.

## Primary colon fibroblast isolation and culture experiments

For fibroblast cultures, normal colon fragments after epithelial crypt isolation or minced AOM/DSS tumors were washed in PBS and

further incubated with 10 mM EDTA. After 1 h, tissue fragments were seeded into 10-cm culture plates coated with 0.1% gelatine (Sigma, G2500) in DMEM (Thermo Fisher Scientific, 31966-047) with 10% FBS (Sigma, F7524) and Pen/Strep (Thermo Fisher Scientific, 15140-122). After several days, fibroblasts expanded from tissue fragments attached to the plates, which were collected by trypsinization and propagated as regular two-dimensional cell cultures on gelatine-coated plates. Rates of proliferation were analyzed by plating 200 fibroblasts in hexaplicates in 96-wells grown for 10 days, respectively, and cell confluence was determined by live-cell microscopy using an IncuCyte S3 instrument (Sartorius). Cell senescence was determined by the Senescence β-Galactosidase Staining Kit (Cell Signaling, 9860) 3 days after plating $1 \times 10^4$ fibroblasts in six-well plates according to the manufacturer's instructions. SA-β-Gal positive and total cell numbers were counted from 15 random images each (>700 cells per line).

For the collagen contraction assay, $1 \times 10^5$ fibroblasts were seeded in 500 μL Collagen I solution (Thermo Fisher Scientific, A1048301) adjusted to 1 mg/mL according to the manufacturer's instructions into a 24-well suspension plate. After polymerization of collagen, the collagen disks were detached from the plate and contraction was determined by monitoring the area of the discs over time.

For the barrier function and T-cell attraction assays, splenocytes were isolated from OT1 mice (Hogquist et al, 1994). T cells were stimulated by the addition of SIINFEKL peptide (AnaSpec, AS-60193-1) for 4 h and expanded for 3 days prior to an experiment. For the barrier function assay, 12-well transwells with 8-μm pore size (Greiner, 665638) were coated with 0.1% gelatine, and $2 \times 10^5$ fibroblasts were seeded on top of the transwell. After 48 h, $3 \times 10^4$ T cells were added on top of the confluent fibroblast layer and cells in the lower compartment were monitored microscopically. For the T-cell attraction assay, $1 \times 10^5$ fibroblasts were seeded into a 12-well adhesion culture plate. After 24 h, a transwell with 3-μm pore size (Corning, CLS3462-48EA) containing $3 \times 10^4$ T cells was added to the well, and attracted cells in the lower compartment were monitored microscopically.

For inflammatory activation, colon fibroblasts were plated in a six-well plate and after 24 h treated with 1 ng/mL murine recombinant IL1α (Biolegend, 575002) in PBS for the indicated periods. The medium was replaced by 1 mL of fresh medium 2 h before treatment and supplemented with 1 mL 2 ng/mL IL1α containing medium at the starting point. At indicated time points, cells were harvested and processed for RNA and protein isolation.

For analysis in 3D, $2 \times 10^5$ fibroblasts were seeded in $8 \times 25$ μL of BME and cultured in Advanced DMEM/F12. Cells were analyzed either untreated or treated with 10 ng/mL recombinant hTGFβ (Peprotech, 100-21) or 1 ng/mL recombinant mIL1α. Medium was replaced by fresh medium every 24 h and cells were harvested and processed for RNA isolation after 96 h.

## Histology, immunohistochemistry (IHC), and immunofluorescence (IF)

Tumors of orthotopic transplantation were dissected and fixed in 4% PFA/PBS overnight at 4 °C. For histological and IHC analyses, paraffin-embedded tumors were sectioned at 3 μm and stained using the Bond-Max device (Leica) and the Bond Polymer Refine Detection system (Leica, DS9800). Sections were imaged using an Aperio CS2 digital pathology slide scanner (Leica) and marker

expression was quantified with macros in Aperio ImageScope (v12.4).

AOM/DSS and AOM/p53 colons were collected, flushed with PBS and longitudinally opened for imaging and determining tumor sizes and numbers using a caliper. Colons were mounted as Swiss Rolls, fixed in 4% PFA/PBS overnight at 4 °C, paraffin-embedded, sectioned at 3–4 μm and subjected to hematoxylin/eosin (H&E), IHC or IF staining as described previously (Krebs et al, 2017). After antibody incubations, washing steps, and DAB reactions, the slides were counterstained with Mayer's hematoxylin before dehydration and mounting (Roti®-Histokitt, 6638.2). Analysis and image acquisition was performed using a Leica DM5500B microscope. Scoring of infiltration into AOM/DSS tumors was estimated from the cellularities of infiltrated cell-type marker-positive cells on IHC slides, with none (0), rare (1), few (2), several (3), and many/abundant (4) classification. For anti-ZEB1 IF of Fib$^{Ctrl}$/Fib$^{ΔZeb1}$ (Col6a1-Cre) mice, cryosections from fresh frozen colon specimens were fixed in 4% PFA for 10 min, permeabilized for 10 min in 0.25% Triton X-100/PBS, blocked in 3% BSA/PBS and incubated with anti-ZEB1 antibodies. After washing and incubation with Alexa594-conjugated secondary antibodies, DAPI-stained sections were mounted (Antifadent AF1, Citifluor). Images were acquired using a Leica DM5500B microscope.

For immunofluorescence staining of tumors from orthotopic transplantation, citrate-based antigen retrieval was applied with deparaffinized sections. Slides were blocked with 20% goat serum (Merck, G9023) in PBST and stained for 1 h with primary antibodies and for 30 min with secondary Alexa488/647-conjugated antibodies. Sections were imaged using an Evos FL microscope and marker expression was quantified using CellProfiler (McQuin et al, 2018). Antibodies and dilutions are listed in Appendix Table S2.

## Multiplex immunofluorescence analysis

FFPE sections (3 μm) from orthotopic tumors (AKP) were stained with Opal 7-Color Automation IHC Kits (Akoya Bioscience) in the BOND-RX Multiplex IHC Stainer (Leica) following established protocols (Strack et al, 2020). Each section was put through 6 sequential rounds of staining, which included blocking in 5% BSA followed by incubation with primary antibodies; for staining conditions see Appendix Table S2. Nuclei were counterstained with 4′,6-diamidino-2-phenylindole (DAPI) contained in the Opal 7-Color Automation IHC Kits and slides were mounted with Fluoromount-G (SouthernBiotech). Imaging was performed with the PhenoImager HT imaging system (Akoya Bioscience). Tumor regions of interest were defined manually, and out-of-focus images or high background signals were excluded from further analysis. Images were analyzed using the phenotyping application of the 'inForm' software V2.54.10 (Akoya Bioscience) and the fluorescence intensities for individual cells were exported. Epithelial cells ($33 \pm 6\%$) and immune cells ($12 \pm 5\%$) were excluded by EPCAM or CD45 intensity thresholds, respectively. CAFs ($7 \pm 2\%$) were separated from other cells by thresholds of αSMA, C3 or MHCII. Assigned phenotypes were myCAFs (αSMA+, C3−, MHCII−), iCAFs (C3+, αSMA−, MHCII−), apCAFs (MHCII+, αSMA−, C3−) and mixed myCAF/iCAF identity (αSMA+, C3+, MHCII−). Cell numbers were then exported and statistically analyzed by Student's $t$ test. Normalized intensity values of αSMA, C3 and MHCII were used to plot CAFs by

UMAP (umap V0.2.10.0) and to display the staining intensities and the cell density distribution (*ndensity*, ggplot2 V3.4.2).

## Picrosirius red staining

Slides were deparaffinized, rehydrated, and then incubated in Picrosirius Red solution (abcam, ab246832) for 1 h at RT before washing in 0.5% acidified water, dehydration, and mounting (Roti®-Histokitt, 6638.2). Polarized light imaging was done by using a Leica DM5500B microscope equipped with a polarization filter to monitor green thinner collagen fibers and yellow-orange bundled fibers by refraction and birefringence. Percentages of total stained areas were analyzed on polarized light images using CellProfiler by RGB conversion, global background-based thresholding for pixel identification, followed by binary image generation of each channel for measuring area occupied by pixels.

## Western blot analysis

Protein extraction and western blotting were carried out as described (Krebs et al, 2017; Schuhwerk et al, 2022). Briefly, cells grown in 6-well plates were lysed in 150 mM NaCl, 50 mM Tris-HCl pH 8.0, 0.5% Na-Desoxycholate (w/v), 0.1% SDS (w/v), 1% NP40 (v/v), 1× complete protease inhibitor (Roche, 4693132001), 1 mM PMSF, 1× PhosSTOP (Roche, 4906837001) for 20 min at 4 °C. Protein concentrations were determined by using the BCA Protein Assay (Thermo Fisher Scientific, 23225) according to manufacturer's instructions. Up to 30 µg of protein lysate was separated by SDS-PAGE and transferred to nitrocellulose membranes before antibody incubation. Detection was carried out using Western Lightning Plus-ECL (Perkin Elmer, NEL103001EA) and a ChemiDoc™ Imaging System (BioRad). Antibodies and dilutions are listed in Appendix Table S2.

## Secretome analysis

For secretome analysis, a commercially available secretome array kit was used (R&D systems, ARY028) according to the manufacturer's instructions. Samples from AOM/DSS CAF supernatants were prepared by surgically detaching adenomas and mincing using scalpels and incubating the tissue in digestion buffer containing 0.05% Collagenase D (w/v), 0.3% Dispase II (w/v), 0.05% DNase I (w/v), 4% FBS (v/v), in DMEM/F12 medium (Thermo Fisher Scientific, 31331028) for 30-45 min at 37 °C with constant agitation. In total, 10 mL of cold washing buffer (sterile PBS containing 2% FBS) was added, and the suspension passed through a 70 µm strainer. After centrifugation and erythrocyte lysis in ACK buffer (150 mM $NH_4Cl$, 10 mM $KHCO_3$, 0.1 mM EDTA; pH 7.2-7.4) for 2 min at RT, washed and collected cells were resuspended in DMEM/F12/10% FBS and plated in 12-well plates. Adherent cells were passaged in 1:1 ratios when reaching confluence. After 2–3 passages, the supernatants of individual confluent 6-well vessels were collected and kept on ice. After centrifugation at $1000 \times g$ for 5 min at 4 °C, supernatants were aliquoted and frozen at −80 °C. For the secretome array, samples were added to the equilibrated and blocked membranes for incubation overnight at 4 °C. After incubation with the detection antibody cocktail for 1 h, with the Streptavidin-HRP mix for 30 min and the detection mix for 1 min

at RT, secretomes were detected by imaging like for western blot. Intensities were normalized to the background of each respective membrane.

## RNA isolation and quantitative reverse transcriptase (qRT-)PCR

Total RNA of cultured cells was isolated using the RNeasy Plus Mini Kit (Qiagen, 74136) or the NucleoSpin RNA kit (Macherey-Nagel, 740955.250) and 200–500 ng of total RNA was used to synthesize cDNA using the RevertAid First Strand cDNA Synthesis Kit (Thermo Fisher Scientific, K1622) or M-MLV Reverse Transcriptase, RNase H Minus, Point Mutant (Promega, M3682) according to manufacturers' instructions. Subsequent qRT-PCR was performed in triplicates in 384-well plates using primers and Roche universal probe library (UPL) with TaqMan™ Universal MasterMix II (Thermo Fisher, 4440044) and LightCycler® 480 II (Roche) or in 96-well plates using primers with PowerUp™ SYBR™ Green MasterMix (Thermo Fisher, A25778) and CFX Opus 96 Real-Time PCR System (Biorad). Primer sequences are listed in Appendix Table S3.

## Single-cell RNA sequencing (scRNA-seq) and analysis

Single-cell transcriptomes were generated using a commercially available 384-well plate approach (SORT-Seq2, Single Cell Discoveries, SCD (Muraro et al, 2016)). To this end, primary tumors were dissected and dissociated into single cells as for establishment of organoid lines from tumors. After erythrocyte lysis in ACK buffer, cells were incubated with Fc Block and antibodies (concentrations listed in Appendix Table S2) in PBS/2% FCS/2 mM EDTA. Single viable cells (eFluor-) were gated for EPCAM+ or CD45+ or double negative cells. EPCAM−, CD45− cells were further gated for CD31 expression and triple-negative cells were considered fibroblasts for sequencing. Cells were index-sorted into 384-well capture plates containing 50 µL lysis buffer and barcoded primers covered by 10 µL of mineral oil by flow cytometry using a BD FACSAria Fusion sorter (BD biosciences) and capture plates were sent for paired-end sequencing at SCD (Illumina Nextseq™ 500). Sequences from read 1 were used for assigning reads to cells and libraries, whereas read 2 was aligned to the ensemble transcriptome (genome assembly GRCm38) using *bwa* version 0.7.10 (Li and Durbin, 2010). The transcript count table was generated by SCD using a custom-written script (https://github.com/anna-alemany/transcriptomics/tree/master/mapandgo). Transcript counts and metadata for all samples were imported and stored as a single-cell experiment object (*SingleCellExperiment* (Amezquita et al, 2020) version 1.12.0). During quality control, cells with high mitochondrial content (*isOutlier*, scater (McCarthy et al, 2017) version 1.18.5) were discarded to remove low-quality cells that may have been damaged during processing or may not have been fully captured by the sequencing protocol. For analysis of CAFs, cells with high expression of *Ptprc* and *Epcam* genes were excluded to avoid contamination by residual immune/epithelial cells. The samples were normalized by computing the $log_2$-transformed normalized expression values across all genes for each cell (*logNormCounts*, scater). Next, all batches were subset to the common features across all samples to enable downstream analysis. To account for the differences in samples due to plates

("batch effect"), we used the fastMNN algorithm (*correctExperiments*, batchelor (Haghverdi et al, 2018) version 1.6.2). Subsequently, clusters were identified using a graph-based approach (*buildSNNGraph* with k set to 15, scran (Lun et al, 2016) version 1.18.5) and the walktrap algorithm (igraph version 1.2.6). Cluster-specific marker genes were identified using the *findMarkers* function (scran) that uniquely define one cluster against the rest. Functional enrichment analysis in Metascape (https://metascape.org/) and Enrichr (https://maayanlab.cloud/Enrichr/) was applied to identify pathways and processes that were enriched in each cluster based on differentially expressed genes (FDR ≤ 0.1; $P \le 0.05$). Furthermore, expression profiles of identified clusters were compared with those previously published for CAF subtypes using the *AddModuleScore* function (Tirosh et al, 2016) from Seurat (Hao et al, 2021) version 4.0.0. To this end, each cell was assigned a score using the module of genes associated with the "published clusters". A positive score suggested that this module of genes is expressed in a particular cell more than would be expected, given the average expression of this module across the population. A mean score for cluster-specific cells was calculated to obtain scores per cluster for each module of genes associated with "published clusters". Using a reference dataset with known labels, the SingleR (Aran et al, 2019) approach (version 1.4.1) labels new cells from a test dataset based on similarity to the reference. Thereby, we assessed the similarity between clusters from Fib$^{\Delta Zeb1}$ and Fib$^{Ctrl}$, where Fib$^{Ctrl}$ was set as a reference. For the scRNA-seq dataset from non-inflammation-driven orthotopic tumors, lower quality clusters with indistinct or ambiguous cell-type identities were excluded by further sub-clustering.

The DAseq tool (Zhao et al, 2021) was used to study the differential abundance of cells from AOM/DSS scRNA Fib$^{Ctrl}$ and Fib$^{\Delta Zeb1}$, using read counts after basic quality control filtering implemented in the standard Seurat (Hao et al, 2021) workflow. Cells with 100–6000 detected genes and less than 10% mitochondrial counts were included. The entire dataset which includes Fib$^{Ctrl}$ and Fib$^{\Delta Zeb1}$ samples, was used to normalize the data, identify the highly variable genes, scale the data, compute the PCA, use nearest neighbor embedding, integrate the batches using harmony and recluster the data. Cell phenotypes were assigned by using Seurat's *AddModuleScore* and published gene sets (Bartoschek et al, 2018; Elyada et al, 2019). DAseq was applied to identify differential abundance (Zhao et al, 2021) and three differentially abundant cell populations were identified. One contained almost exclusively cells from one SORT-Seq plate and was therefore neglected. Statistical significance is reported in the respective figures or figure legends. Characteristic genes for the two other DA populations were identified using the marker gene function included in the DAseq package, which uses stochastic gates.

## Bulk RNA-Seq and GSEA

The integrity of total RNA samples was assessed by Bioanalyzer2100 before sequencing and processed at Novogene UK, Cambridge. RNA-Seq raw reads files were analyzed using nfcore RNA-seq pipeline version 3.11.0 (Ewels et al, 2020). Briefly, raw sequencing reads were aligned to the mouse genome GRCm38 using STAR (Dobin et al, 2013), the reads aligning to each annotated gene were quantified by the Salmon software (Patro et al, 2017), and differential gene expression was performed using

DESeq2 R package (Love et al, 2014). Gene set enrichment analysis (GSEA) was performed on all (filtered) genes across the different experimental conditions using the R packages "clusterProfiler" (v4.10.0) and "enrichPlot" (v1.22.0) with MSigDB and custom gene sets (Elyada et al, 2019; Ohlund et al, 2017; Wu et al, 2021). To account for multiple comparisons, *P* values were adjusted using the Benjamini–Hochberg method.

## Statistics and reproducibility

Information on the number of biologically independent samples analyzed and the number of times experiments were performed is included in the figure legends. All the representative experiments were repeated at least three times unless otherwise stated. Statistical analyses were performed using GraphPad Prism, within the provided R packages or online tools for gene set enrichment analyses, and details on statistical methods are included in the figure legends. All error bars represent SD or SEM as indicated. Where possible animals were assigned to experimental groups using simple randomization and the investigators were blinded for initial animal data collection on tumor/metastasis parameters. No blinding was applied for other experiments since investigators needed information about the groups to correctly perform and analyze the experiments.

# Data availability

scRNA-seq and bulk RNA-seq data have been deposited to Gene Expression Omnibus (GEO) database with the accession codes GSE253368, GSE253639 and GSE253546. All original codes have been deposited at GitHub (bulk RNA-Seq: https://github.com/AG-Stemmler/Menche-Schuhwerk-et-al.-Zeb1_CRC_Manuscript; scRNA-Seq: https://github.com/CUBiDA/zeb1_crc_manuscript).

The source data of this paper are collected in the following database record: biostudies:S-SCDT-10_1038-S44319-024-00186-7.

# Peer review information

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

## Acknowledgements

The authors thank Britta Schlund, Eva Bauer, Friederike Gräbner, Tahmineh Darvishi, Margarete Mijatovic, and Petra Dinse for excellent technical assistance. The authors are grateful to Stefan Stein and Annette Trzmiel (Core Facility Flow Cytometry GSH Frankfurt) for support with all fluorescence-activated cell sorting (FACS) and the animal facilities at FAU Erlangen and GSH Frankfurt for excellent animal care and welfare. The authors thank all members from the Farin, Stemmler, and Brabletz laboratories for critical input and fruitful discussions. This work was supported by the German Research Foundation (FOR2438/P04, TRR305 TP A03, A04, B01 and B07, and BR1399/9-1, BR1399/10-1, BR1399/15-1, BR1399/17-1, BR4145/1-1, BR4145/2-1 and BR4145/3-1), the Wilhelm Sander-Stiftung (2020.039.1), IZKF-Erlangen (IZKF P34, P133 and D39) and the European Union's Horizon 2020 research and innovation program under the Marie Skłodowska-Curie grant agreement (No. 861196, PRECODE). AH and DBB were supported by the German Federal Ministry of Education and Research (BMBF) within the framework of the CompLS funding concept [031L0309A (NetMap)].

## Author contributions

**Constantin Menche**: Formal analysis; Validation; Investigation; Visualization; Methodology; Writing—original draft; Writing—review and editing. **Harald Schuhwerk**: Conceptualization; Formal analysis; Funding acquisition; Validation; Investigation; Visualization; Methodology; Writing—original draft; Writing—review and editing. **Isabell Armstark**: Investigation. **Pooja Gupta**: Data curation; Software; Formal analysis; Investigation; Visualization; Methodology. **Kathrin Fuchs**: Investigation. **Ruthger van Roey**: Data curation; Investigation. **Mohammed H Mosa**: Data curation; Investigation. **Anne Hartebrodt**: Software; Formal analysis; Investigation; Visualization; Methodology. **Yussuf Hajjaj**: Data curation; Software; Investigation. **Ana Clavel Ezquerra**: Investigation. **Manoj K Selvaraju**: Data curation; Software; Investigation; Visualization. **Carol I Geppert**: Resources. **Stefanie Bärthel**: Resources. **Dieter Saur**: Resources. **Florian R Greten**: Resources; Supervision. **Simone Brabletz**: Funding acquisition; Investigation. **David B Blumenthal**: Software; Formal analysis; Supervision; Funding acquisition; Investigation; Visualization. **Andreas Weigert**: Investigation; Visualization; Methodology. **Thomas Brabletz**: Conceptualization; Supervision; Funding acquisition; Investigation; Writing—original draft; Writing—review and editing. **Henner F Farin**: Conceptualization; Formal analysis; Supervision; Funding acquisition; Investigation; Visualization; Writing—original draft; Writing—review and editing. **Marc P Stemmler**: Conceptualization; Formal analysis; Supervision; Funding acquisition; Investigation; Visualization; Writing—original draft; Project administration; Writing—review and editing.

Source data underlying figure panels in this paper may have individual authorship assigned. Where available, figure panel/source data authorship is listed in the following database record: biostudies:S-SCDT-10_1038-S44319-024-00186-7.

## Funding

## Disclosure and competing interests statement

The authors declare no competing interests. Thomas Brabletz is a member of the Advisory Editorial Board of EMBO reports. This has no bearing on the editorial consideration of this article for publication.

# Expanded View Figures

**Figure EV1. Loss of *Zeb1* in fibroblasts does not affect morphology of primary tumors in the orthotopic transplantation model.**

(A) IF images and quantification of ZEB1 expression in tumor stroma after orthotopic transplantation of AKP tumor organoids ($n = 10/9$ independent mice for Fib$^{Ctrl}$/Fib$^{\Delta Zeb1}$, $P < 0.0001$, Student's $t$ test). (B) Quantification of primary tumor engraftment in treatment-naïve Fib$^{Ctrl}$ and Fib$^{\Delta Zeb1}$ mice after orthotopic transplantation of AKP tumor organoids. Numbers of experimental mice are indicated. (C) Tumor volume after orthotopic transplantation of AKP tumor organoids ($n = 8/5$ independent mice for Fib$^{Ctrl}$/Fib$^{\Delta Zeb1}$, $P = 0.7584$, Student's $t$ test). (D, E) Representative H&E stainings of AKP (D) and AKP$^{re}$ (E) tumor sections. Top left corners show higher magnification of the indicated regions. (F–H) Analysis of tumors after orthotopic transplantation of AKP$^{re}$ tumor organoids in treatment-naïve Fib$^{Ctrl}$ and Fib$^{\Delta Zeb1}$ mice. (F) Tumor onset ($n = 4/5$ for Fib$^{Ctrl}$/Fib$^{\Delta Zeb1}$; $P = 0.6401$, Mantel–Cox test). (G) Quantification of tumor engraftment. Numbers of experimental mice are indicated. (H) Tumor volume ($n = 4/4$ independent mice for Fib$^{Ctrl}$/Fib$^{\Delta Zeb1}$). (I–K) Analysis of tumors after orthotopic transplantation of AKP$^{re}$ tumor organoids in control IgG-treated Fib$^{Ctrl}$ and Fib$^{\Delta Zeb1}$ mice. These mice are shown again as controls in Fig. 5. (I, J) Tumor onset (I) and quantification (J) after orthotopic transplantation of AKP$^{re}$ tumor organoids ($n = 12/14$ for Fib$^{Ctrl}$/Fib$^{\Delta Zeb1}$, $P = 0.4564$, Mantel–Cox test). (K) Tumor volumes ($n = 8/10$ independent mice for Fib$^{Ctrl}$/Fib$^{\Delta Zeb1}$). Only tumors collected after day 28 were included. Data information: Data are represented as mean ± SD (A, C, H, K). Scale bars represent 50 μm (A) or 1 mm (D, E). Source data are available online for this figure.

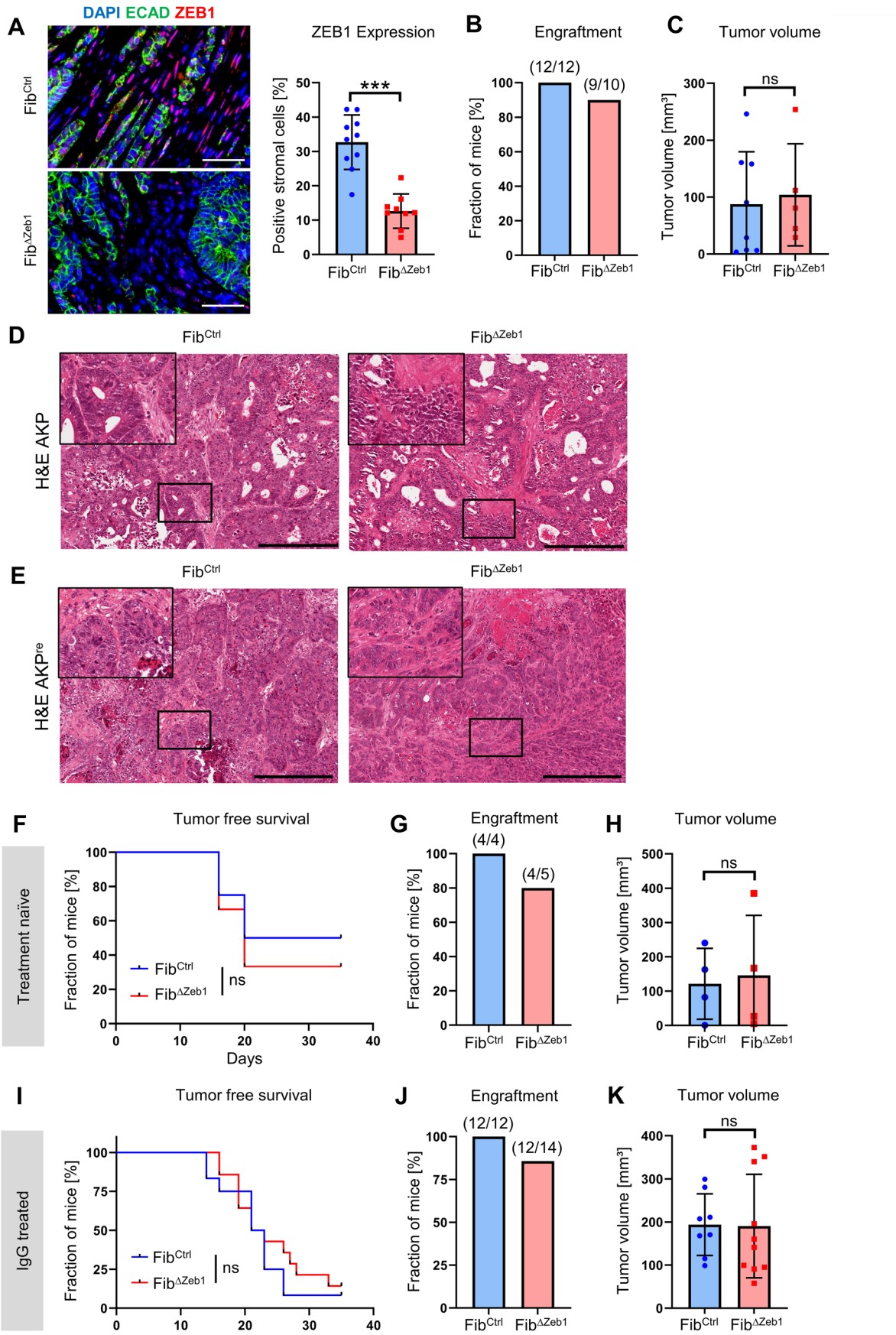

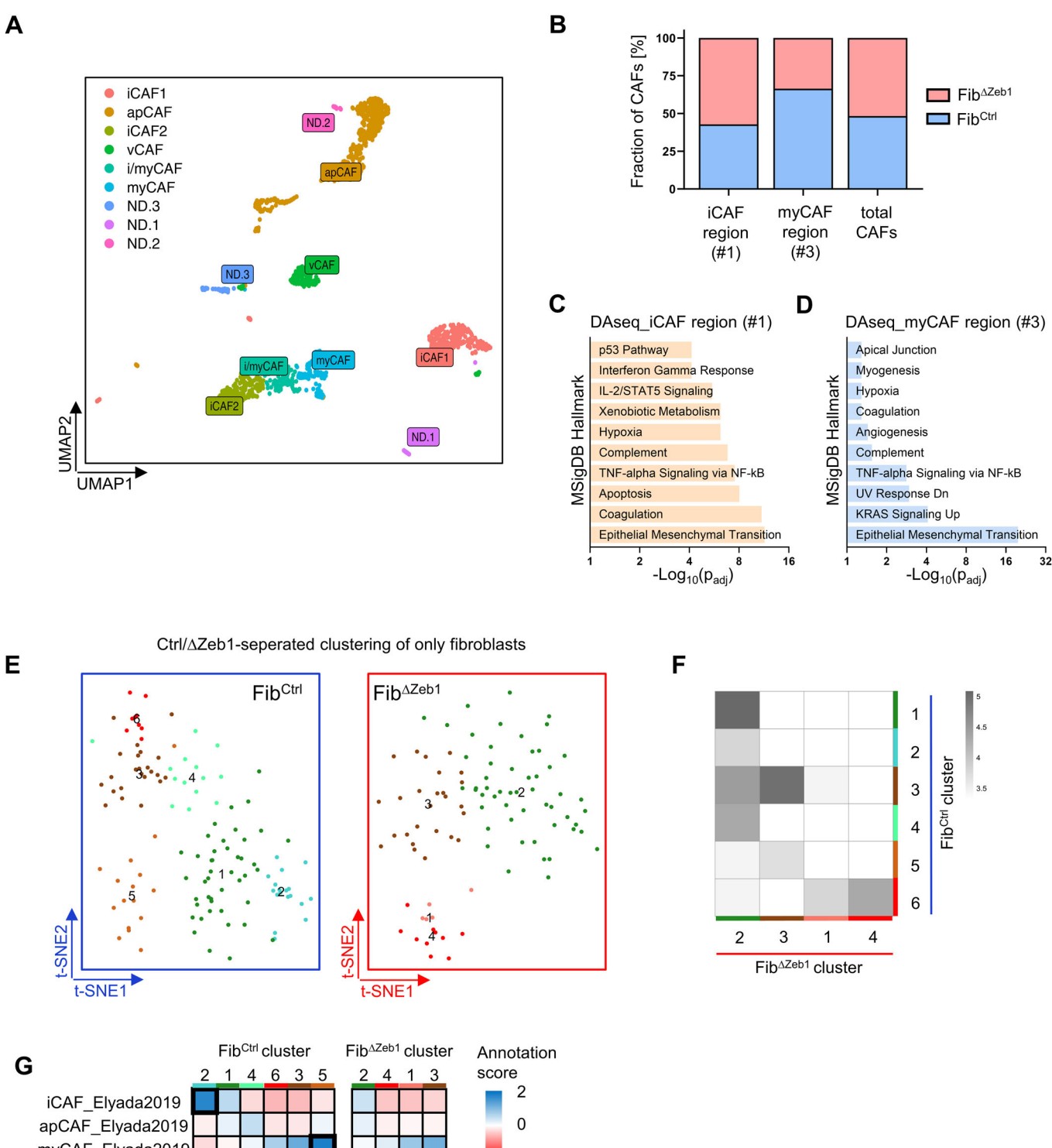

**Figure EV2.  scRNA-seq in AOM/DSS and orthotopic models show reduced CAF diversity and impaired subtype-specific gene expression in Fib^ΔZeb1 tumors.**

(A) Phenotypic annotation of AOM/DSS CAFs ($n = 3$ mice per genotype; corresponding to Fig. 2) upon integrated clustering by scoring of iCAF/myCAF/apCAF/mCAF/vCAF gene signatures (Bartoschek et al, 2018; Elyada et al, 2019) and displayed on UMAP Leiden clusters, pooled according to the determined scores. Note that 'i/myCAFs' share features of iCAFs and myCAFs and that the identities of 3 cell clusters could not be determined (ND) using these gene sets. (B) Genotype distribution of cells in AOM/DSS CAFs in DA regions (please refer to Fig. 2). The fraction of Fib^Ctrl and Fib^ΔZeb1 CAFs in each DA region or among all CAFs is shown. (C, D) Gene set enrichment analysis using Enrichr of marker genes from DAseq region 1 (C) and 3 (D) as compared to all other CAFs (Benjamini-Hochburg corrected Fisher's exact test). (E) t-SNE sub-clustering of fibroblasts separately in Fib^Ctrl (left) and Fib^ΔZeb1 mice (right) of the orthotopic model showing less clusters in Fib^ΔZeb1. (F) Cluster similarities defined by cluster annotations based on 'SingleR' scores (see "Methods" for details) (Aran et al, 2019). Grayscale shows the $\log_2$-transformed number of cells across clusters. Note the low similarity of Fib^ΔZeb1 cells with Fib^Ctrl clusters 2 and 5. (G) Heatmap showing the similarity (annotation scores) of gene expression in CAF clusters with published gene sets. Note, the high scores of Fib^Ctrl clusters 2 and 5 when compared with 'iCAF' and 'myCAF' signatures, respectively, and absence in Fib^ΔZeb1. Source data are available online for this figure.

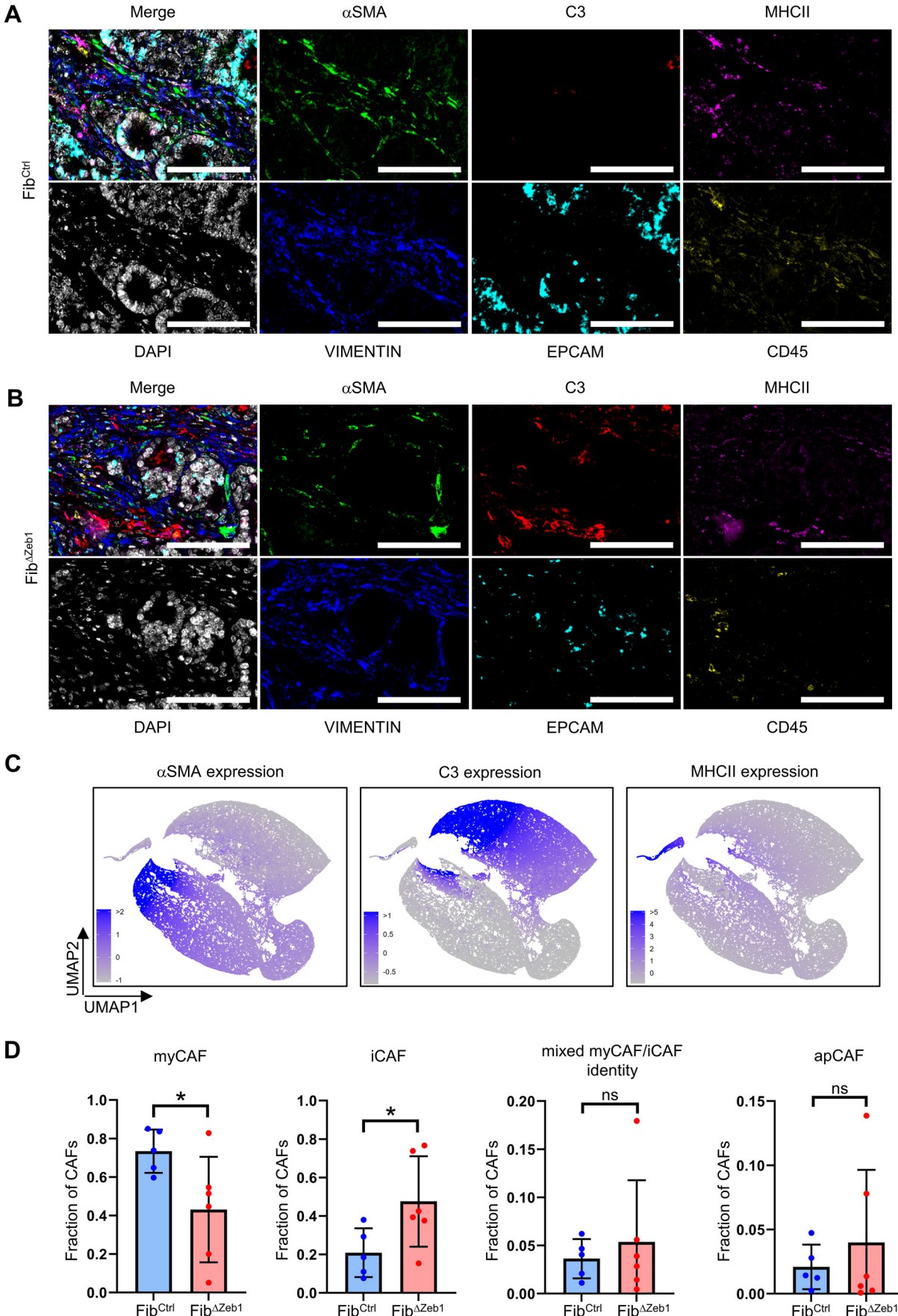

**Figure EV3. Multiplexed IF staining of tumor sections reveals an enrichment of iCAF-like cells in Fib$^{\Delta Zeb1}$ tumors.**

(A, B) Single marker images (VIM, αSMA, C3, MHCII, EPCAM, CD45, DAPI) and merge of all channels for the representative images of Fib$^{Ctrl}$ (A) and Fib$^{\Delta Zeb1}$ (B) orthotopic tumors shown in Fig. 2K. (C) Normalized staining intensities of CAF markers (αSMA, C3, MHCII) in UMAP embedding of CAFs from Fib$^{Ctrl}$ and Fib$^{\Delta Zeb1}$ tumors. (D) Quantification of the distribution of CAF subtypes based on thresholds for αSMA (myCAF-like), C3 (iCAF-like) or MHCII (apCAF) staining intensity ($n = 5/6$ independent mice for Fib$^{Ctrl}$/Fib$^{\Delta Zeb1}$, myCAF: $P = 0.0469$, iCAF: $P = 0.0499$, Student's $t$ test). Data information: Data are presented as mean ± SD (D). Scale bars represent 100 μm (A, B). Source data are available online for this figure.

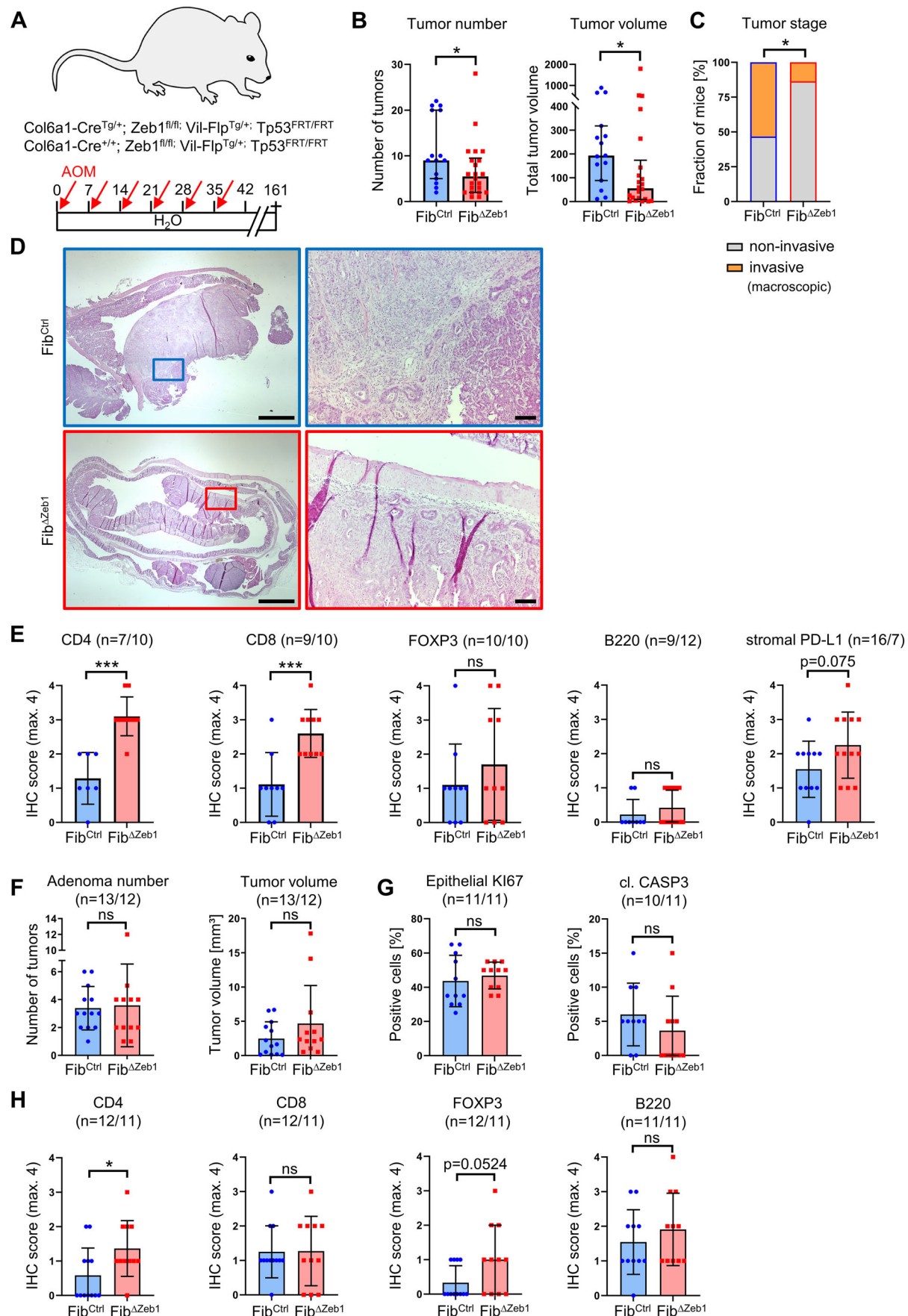

◀ **Figure EV4.   Loss of *Zeb1* in fibroblasts impairs tumor progression and increases T-cell infiltration in the invasive non-inflammation-driven AOM/p53 model.**

(A) Schematic representation of the AOM/p53 model. (B) Quantification of numbers and volumes of tumors in the colons of Fib^Ctrl and Fib^ΔZeb1 mice at the endpoint ($n = 15/22$ for Fib^Ctrl/ Fib^ΔZeb1, number: $P = 0.0433$, volume: $P = 0.0202$, Mann–Whitney test). (C) Macroscopic evaluation of the most advanced/progressed tumor per mouse categorizing 'T4' as fully invasive (penetrating the muscle) or not ('T1-T3') (fraction of mice is given, $n = 15/22$ for Fib^Ctrl/Fib^ΔZeb1, $P = 0.0252$, Fisher's exact test). (D) Representative H&E stainings with a higher magnification of the indicated region to the right. (E) IHC-based quantification of immune cell infiltration and stromal PD-L1 expression of tumors from Fib^Ctrl and Fib^ΔZeb1 mice. Numbers of experimental mice per genotype are indicated (CD4, CD8, FOXP3, B220, PD-L1: $P < 0.0001$, $P = 0.0010$, 0.3618, 0.3748, 0.0745, Student's *t* test). (F–H) AOM/DSS model until day 50 in Fib^Ctrl and Fib^ΔZeb1 mice. Macroscopic analysis of early adenomas (number: $P = 0.8339$, volume: $P = 0.2088$, Student's *t* test) (F) and IHC quantification of epithelial proliferation and cell death (KI67: $P = 0.5403$, cl. CASP3: $P = 0.2773$, Student's *t* test) (G), as well as immune cell infiltration (H). Numbers of experimental mice per genotype are indicated (CD4, CD8, FOXP3, B220: $P = 0.0296$, 0.9515, 0.0524, 0.3996, Student's *t* test). Data information: Data are presented as mean ± SD (B, E–H). Scale bars represent 1.5 mm (D, left) or 100 μm (D, right). Source data are available online for this figure.

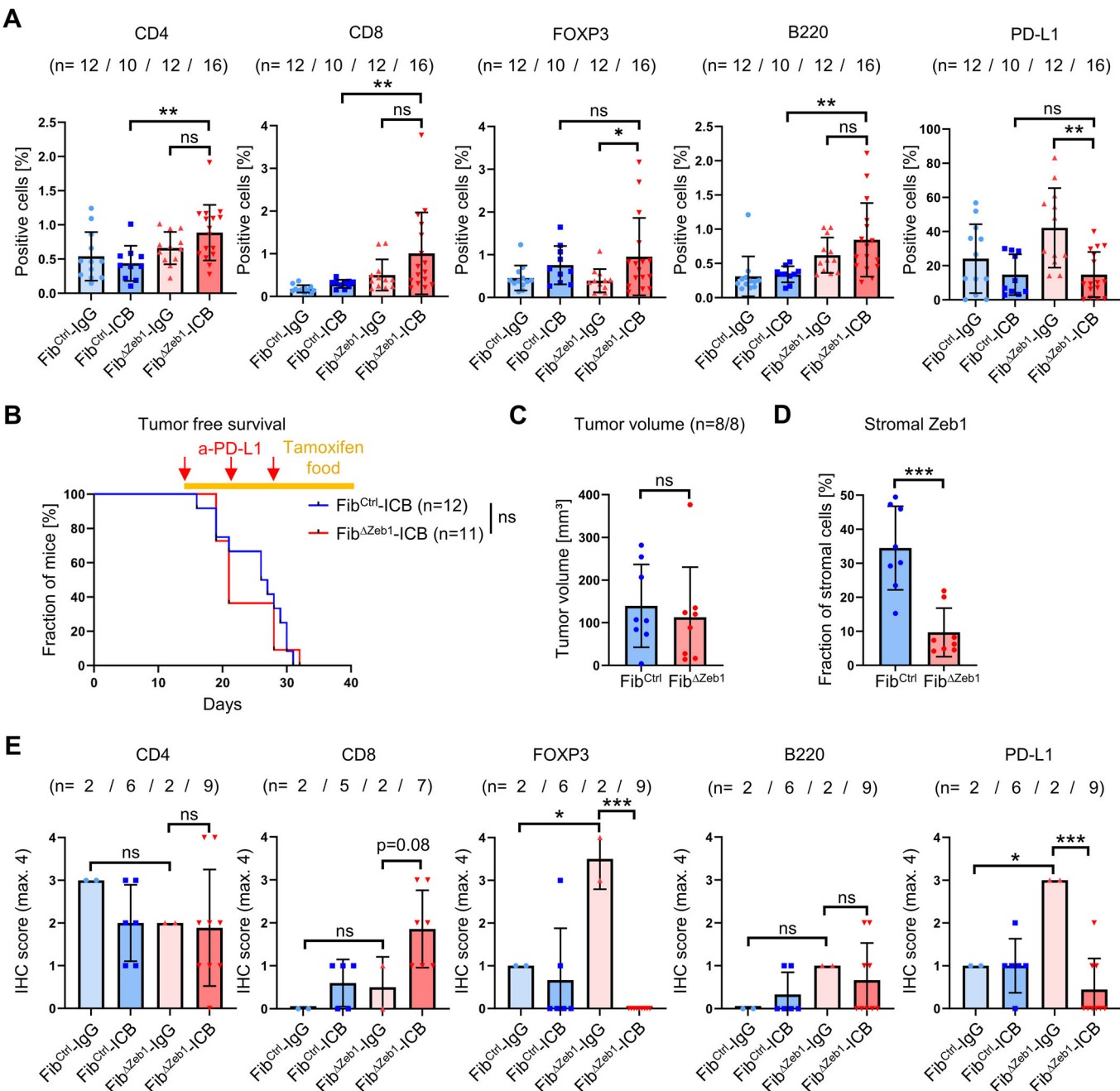

**Figure EV5. Monitoring of immune cell infiltration after ICB and effect of late-stage deletion of *Zeb1* on ICB.**

(A) IHC-based quantification of immune cell infiltration and PD-L1 expression in tumors from Fib^Ctrl and Fib^ΔZeb1 mice after orthotopic transplantation of AKP^re organoids and intraperitoneal injection of a-PD-L1 antibodies or control IgGs. Numbers of experimental mice per condition are indicated (CD4, CD8, FOXP3, B220, PD-L1: $P = 0.0032, 0.0076, 0.6442, 0.0027, >0.9999$ (Fib^Ctrl-ICB vs Fib^ΔZeb1-ICB), p = 0.1568, 0.0525, 0.0304, 0.2150, 0.0004 (Fib^ΔZeb1-IgG vs Fib^ΔZeb1-ICB), Šídák's multiple comparisons test). IgG-treated mice contributed to initial AKP^re analysis (Fig. 1). (B) Kaplan–Meier analysis showing tumor-free survival of Fib^Ctrl and Fib^ΔZeb1 mice after orthotopic transplantation of AKP^re organoids and intraperitoneal injection of a-PD-L1 antibodies or control IgGs, as indicated by red arrows. Recombination of *Zeb1* was induced by tamoxifen food starting from day (d) 14. Mice were considered tumor-free if no tumor was detected by palpation. Numbers of experimental mice per condition are indicated ($P = 0.7583$, Mantel–Cox test). (C) Quantification of tumor volumes after orthotopic transplantation of AKP^re tumor organoids and ICB with late recombination of *Zeb1*. Only tumors collected after d28 were included in this analysis ($n = 8$ independent mice, $P = 0.6276$, Student's $t$ test). (D) IF-based quantification of stromal cells expressing ZEB1 ($n = 8/8$ independent mice for Fib^Ctrl/Fib^ΔZeb1, $P = 0.0002$, Student's $t$ test). (E) IHC-based quantification of immune cell infiltration and PD-L1 expression in AOM/DSS tumors from Fib^Ctrl and Fib^ΔZeb1 mice receiving ICB (2 intraperitoneal injections of a-PD-L1 and a-CTLA-4 antibodies or control IgGs per week starting at d70 of AOM/DSS tumorigenesis). Numbers of experimental mice per condition are indicated (CD4, CD8, FOXP3, B220, PD-L1: $P = 0.9902, 0.0802, <0.0001, 0.0070, 0.0003$ (Fib^ΔZeb1-IgG vs Fib^ΔZeb1-ICB), $P = 0.6241, 0.7615, 0.0070, 0.3163, 0.0144$ (Fib^Ctrl-IgG vs Fib^ΔZeb1-IgG), Šídák's multiple comparisons test). Data information: Data are presented as mean ± SD (A, C–E). Source data are available online for this figure.

