## [Peer Review File · EMBO Reports]

ZEB1-mediated fibroblast polarization controls inflammation and sensitivity to immunotherapy in colorectal cancer

Constantin Menche, Harald Schuhwerk, Isabell Armstark, Pooja Gupta, Kathrin Fuchs, Ruthger van Roey, Mohammed Mosa, Anne Hartebrodt, Yussuf Hajjaj, Ana Clavel Ezquerra, Manoj Selvaraju, Carol Geppert, Stefanie Bärthel, Dieter Saur, Florian Greten, Simone Brabletz, David Blumenthal, Andreas Weigert, Thomas Brabletz, Henner Farin, and Marc Stemmler

Corresponding author(s): Marc Stemmler (marc.stemmler@fau.de), Thomas Brabletz (thomas.brabletz@fau.de), Henner Farin (h.farin@georg-speyer-haus.de)

Review Timeline:

Submission Date:	16th Apr 24
Editorial Decision:	24th Apr 24
Revision Received:	21st May 24
Editorial Decision:	3rd Jun 24
Revision Received:	10th Jun 24
Accepted:	13th Jun 24

Editor: Achim Breiling

Transaction Report: *This manuscript (previously peer reviewed and revised at a journal outside EMBO press) was transferred to EMBO reports following arbitration at The EMBO Journal.*

Dear Dr. Stemmler,

Thank you for transferring your manuscript to EMBO reports. I now went again through the manuscript and the reports of the arbitrators from The EMBO Journal (attached again below).

Arbitrator #2 has some concerns and suggestions to improve the manuscript, I ask you to address in a final revised manuscript. Please revise your manuscript with the understanding that all concerns of the arbitrator must be addressed in the revised manuscript or in a detailed point-by-point response. As discussed in our video call, please provide further explanations and critical discussion regarding and clarify the description of the data mentioned by the arbitrator. However, we do not require that these parts of the manuscript ('claims on tumor growth, the combination with CPI and enthusiasm of therapeutic use') will be removed.

PLEASE NOTE THAT upon resubmission revised manuscripts are subjected to an initial quality control. Upon failure in the initial quality control, the manuscripts are sent back to the authors, which may lead to delays. Frequent reasons for such a failure are the lack of the data availability section (please see below) and the presence of statistics based on $n=2$ (the authors are then asked to present scatter plots or provide more data points).

1) a .docx formatted version of the final manuscript text (including legends for main figures, EV figures and tables), but without the figures included. Please make sure that changes are highlighted to be clearly visible. Figure legends should be compiled at the end of the manuscript text.

2) individual production quality figure files as .eps, .tif, .jpg (one file per figure), of main figures and EV figures. Please upload these as separate, individual files upon re-submission. Please make sure that all figure panels are called out separately and sequentially in the manuscript text

For more details please refer to our guide to authors:

See also our guide for figure preparation:

Moreover, please consult our guidelines for figure legend preparation:

3) a .docx formatted letter INCLUDING the arbitrators' report(s) and your detailed point-by-point responses to the comments. As part of the EMBO Press transparent editorial process, the point-by-point response is part of the Review Process File (RPF), which will be published alongside your paper.

4) a complete author checklist, which you can download from our author guidelines

(<https://www.embopress.org/page/journal/14693178/authorguide>). Please insert page numbers in the checklist to indicate where the requested information can be found in the manuscript. The completed author checklist will also be part of the RPF.

5) that primary datasets produced in this study (e.g. RNA-seq, ChIP-seq and array data) are deposited in an appropriate public database. This is now mandatory (like the COI statement). If no primary datasets have been deposited in any database, please

state this in this section (e.g. 'No primary datasets have been generated and deposited').

The accession numbers and database should be listed in a formal "Data Availability" section (placed after Materials & Methods) that follows the model below. Please note that the Data Availability Section is restricted to new primary data that are part of this study.

Data availability

8) Regarding data quantification and statistics, please make sure that the number "n" for how many independent experiments were performed, their nature (biological versus technical replicates), the bars and error bars (e.g. SEM, SD) and the test used to calculate p-values is indicated in the respective figure legends (also for potential EV figures and all those in the final Appendix). Please also check that all the p-values are explained in the legend, and that these fit to those shown in the figure. Please provide statistical testing where applicable. Please avoid the phrase 'independent experiment', but clearly state if these were biological or technical replicates. Please also indicate (e.g. with n.s.) if testing was performed, but the differences are not significant. In case n=2, please show the data as separate datapoints without error bars and statistics.

See also:

<http://www.embopress.org/page/journal/14693178/authorguide#statisticalanalysis>

9) Please note our reference format:

10) We updated our journal's competing interests policy in January 2022 and request authors to consider both actual and perceived competing interests. Please review the policy <https://www.embopress.org/competing-interests> and add a statement declaring your competing interests. Please name that section 'Disclosure and Competing Interests Statement' and add it after the author contributions section.

11) Please provide a title with not more than 100 characters (including spaces), add up to five keywords to the manuscript and order the sections like this using these names:

Title page - Abstract (175 words) - Keywords - Introduction - Results - Discussion - Methods - Data availability section (DAS) - Acknowledgements - Disclosure and Competing Interests Statement - References - Figure legends - Expanded View Figure legends

12) Please add scale bars of similar style and thickness to all microscopic images, using clearly visible black or white bars (depending on the background). Please place these in the lower right corner of the images themselves. Please do not write on or near the bars in the image but define the size in the respective figure legend.

13) Please make sure that all the funding information is also entered into the online submission system and is complete and similar to the one in the manuscript text file (in the Acknowledgements).

14) We now use CRediT to specify the contributions of each author in the journal submission system. CRediT replaces the author contribution section. Please use the free text box to provide more detailed descriptions. Thus, please do not provide your final manuscript text file with an author contributions section. See also guide to authors:
<https://www.embopress.org/page/journal/14693178/authorguide#authorshipguidelines>

15) We would encourage you to use 'Structured Methods', our new Materials and Methods format. According to this format, the Materials and Methods section should include a Reagents and Tools Table (listing key reagents, experimental models, software, and relevant equipment and including their sources and relevant identifiers) followed by a Methods and Protocols section in which we encourage the authors to describe their methods using a step-by-step protocol format with bullet points, to facilitate the adoption of the methodologies across labs. More information on how to adhere to this format as well as downloadable templates (.doc or .xls) for the Reagents and Tools Table can be found in our author guidelines (section 'Structured Methods'):

In addition, I would need from you:

I look forward to seeing a revised version of your manuscript when it is ready. Please let me know if you have questions or comments regarding the revision.

Please use this link to submit your revision: <https://embor.msubmit.net/cgi-bin/main.plex>

Kind regards,

Achim

Arbitrator #1:

I have gone through the manuscript and rebuttal letters. My opinion is positive. The data are overall convincing and sustain the authors' conclusions. The results are novel. There have been many studies correlating CAF activity, immune evasion, and poor prognosis in CRC, but none I know shows direct genetic evidence that manipulation of CAF polarization in vivo modulates T infiltration and immunotherapy response. In addition, this study provides experimental in vivo evidence supporting the known dichotomy of CAF phenotypes; MyCAFs driven by TGF-beta (and Zeb1) and iCAFs driven by IL1. I like this paper. It is a relevant contribution.

Arbitrator #2:

I took a close look at the paper and the reviewers comments. I fully agree with your evaluation. I believe that the existing work in breast cancer does not decrease the merit of this study. That paper (ref 25) only reports expression of Zeb1 in breast tumor CAFs. There is no functional experiment. Also, I agree that a full investigation of the mechanism underlying Zeb1 activity in CAFs seems very much outside the scope of this paper. Convincing activity should first be established and this where the challenge lies with the current study.

The issue for me is with the quality of the in vivo studies and the magnitude of the effect on tumorigenesis:

- Whether it is the DSS/AOM or the AKP model, the impact of Zeb1 deletion appears to be minor. I do not think that you can resolve this by crossing to an immune deficient background or depleting T-cells. The effect is so small and variable that this is not going to provide a clean answer.
- The impact on metastasis, if real, would be very hard to explain. It could be just decrease growth of the primary tumor but could be the inability of metastatic cells to initiate growth at the secondary side due to a change in CAF phenotype. In any case, as the effect on primary tumors is so small and variable, I do not think you can claim anything about metastasis.

-The combination experiments with Checkpoint inhibitors are very weak. If any of it was mediated by TGFb, the results would be black and white. This result has been published many time. Here is one example with 70% CR in a CRC model, MC38: (<https://www.ncbi.nlm.nih.gov/pmc/articles/PMC6028240/>).

In short, I would recommend to go with your second option. Minimize or remove any claims on tumor growth, the combination with CPI and enthusiasm of therapeutic use (not that anyone will get excited by Zeb1 as a therapeutic target at this time). Limit the paper to describing the change in polarization of CAFs and potential impact on immune contexture and go with EMBO Report rather than EMBO J.

Reviewers' comments:

We would like to thank both arbitrary advisors for the critical evaluation of our revised manuscript. We are grateful that both arbitrators have acknowledged our work and expressed their general support for a publication in EMBO Reports.

However, Arbitrator 2 still raises concerns on “the quality of the in vivo studies and the magnitude of the effect”. As outlined below we kindly disagree with these points and would like to stress that the unique strength of our work is that we have documented the phenotypic outcomes of CAF manipulation in three independent autochthonous/orthotopic CRC models. Our analysis shows consistent primary defects in all models (i.e. modulation of CAF subtypes, increased inflammation and immune cell infiltration) that differentially affect phenotypic outcomes in a stage and context-specific manner. We are convinced that by conducting such deep phenotypic analysis, which goes far beyond state-of-the-art, we provide a more global picture that improves our understanding on CAF biology in the dynamic tumor microenvironment. These findings have important clinical implications and will help to obtain more informative models for immune checkpoint therapy in CRC.

In a point-by-point response, we address the raised points.

Arbitrator #1:

I have gone through the manuscript and rebuttal letters. My opinion is positive. The data are overall convincing and sustain the authors' conclusions. The results are novel. There have been many studies correlating CAF activity, immune evasion, and poor prognosis in CRC, but none I know shows direct genetic evidence that manipulation of CAF polarization in vivo modulates T infiltration and immunotherapy response. In addition, this study provides experimental in vivo evidence supporting the known dichotomy of CAF phenotypes; MyCAFs driven by TGF-beta (and Zeb1) and iCAFs driven by IL1. I like this paper. It is a relevant contribution

We thank the arbitrator for his/her acknowledgement that our data are convincing and novel and that the paper is a relevant contribution.

Arbitrator #2:

I took a close look at the paper and the reviewers comments. I fully agree with your evaluation. I believe that the existing work in breast cancer does not decrease the merit of this study. That paper (ref 25) only reports expression of Zeb1 in breast tumor CAFs. There is no functional experiment. Also, I agree that a full investigation of the mechanism underlying Zeb1 activity in CAFs seems very much outside the scope of this paper. Convincing activity should first be established and this where the challenge lies with the current study.

We fully agree with the reviewer that the study by Fu et al. does not decrease the merit of our work and that precisely deciphering the molecular mechanism underlying ZEB1 activity in CAFs is outside of the scope of our study.

The issue for me is with the quality of the in vivo studies and the magnitude of the effect on tumorigenesis:

-Whether it is the DSS/AOM or the AKP model, the impact of Zeb1 deletion appears to be minor. I do not think that you can resolve this by crossing to an immune deficient background or depleting T-cells. The effect is so small and variable that this is not going to provide a clean answer.

We kindly disagree: The endpoints, as well as the associated immuno-phenotyping show consistent and significant changes in all 3 in vivo models studied. We therefore think that this criticism must have resulted from a misunderstanding. We have now more clearly explained the individual and common findings between our models (see text passages labelled in red and graphical abstract). Notably, we found increased immune cell infiltration in Fib Δ Zeb1 mice in all 3 models. The observed differences in specific immune cell subsets are likely a result of the fundamentally distinct etiologies of the studied models. In the inflammation-driven tumor model (AOM/DSS) we observed increased adenoma formation (size and number). This is compatible with our hypothesis that the increased inflammatory response in AOM/DSS tumorigenesis in combination with the altered inflammatory CAF subtype upon Zeb1 depletion increases tumor initiation and growth. This idea is further supported by the finding that the primary tumor growth was unaffected in both sporadic models (orthotopic transplantation and AOM/p53). In contrast, the consequences of ZEB1 dependent CAF modulation in advanced stages could only be addressed in the sporadic models because the AOM/DSS tumors remain benign. Here, we could observe that *Zeb1* deletion is associated with decreased local invasion (AOM/p53) and distant metastasis (orthotopic transplantation), indicating that the CAFs modulation can counteract tumor progression. Consistent with the common increase of immune cell infiltration we found that immune checkpoint therapy resulted in reduced tumor burden in both sporadic and inflammation induced tumors. Together, our results highlight how ZEB1 in CAFs drives tumor-TME crosstalk in a context and stage-specific manner, which commonly results in increased immune cell infiltration. Furthermore, we agree that experiments in an immunodeficient background or T-cell depletion would not add substantially new mechanistic insights.

-The impact on metastasis, if real, would be very hard to explain. It could be just decrease growth of the primary tumor but could be the inability of metastatic cells to initiate growth at the secondary side due to a change in CAF phenotype. In any case, as the effect on primary tumors is so small and variable, I do not think you can claim anything about metastasis.

We agree that the exact mechanism how ZEB1 affects tumor progression by modulating CAF composition and immune cell infiltration remains open. However, we are puzzled that the reviewer questions the general relevance of our observations. Notably, we show significant and profound reduction of metastasis incidence and number in spontaneous transplantation models that provide a high physiologic relevance (Fig. 1K, L) Because, we did not observe any parallel reduction of primary tumor growth this points towards a specific anti-metastatic effect. While our data does not exclude an additional role of ZEB1 in CAFs at the secondary site, we show significantly increased T and B-cell infiltration in the primary tumors arguing for increased anti-tumor immunity (Fig. 4B) that was independently observed also in the AOM/p53 model (Fig. EV4). In summary, we are convinced that our accumulated data provide consistent evidence about this key aspect of our analysis indicating that ZEB1 orchestrates CAF diversification and formatting of the TME, which has major impact on tumor biology and immunity.

-The combination experiments with Checkpoint inhibitors are very weak. If any of it was mediated by TGF β , the results would be black and white. This result has been published many times. Here is one example with 70% CR in a CRC model, MC38: (<https://www.ncbi.nlm.nih.gov/pmc/articles/PMC6028240/>).

Given the clear and consistent changes described in Fig. 5 and Fig. EV5 we are surprised that the arbitrator considers our immune checkpoint blockade experiments as “weak”. We think that referring to a study in which 2D cell lines have been subcutaneously injected for

ICB treatment is rather inadequately challenging our results by ignoring that we exclusively used autochthonous and orthotopic transplantation models with high clinical relevance. We also need to stress that MC38 cells are known to be responsive to ICB (hence a 70% CR), which holds true only for a minority of patient's CRC tumors. The importance of our findings is that in our models neither AOM/DSS or sporadic tumors were responsive to ICB, but depleting *Zeb1* from CAFs and the subsequent TME remodeling rendered these tumors responsive to ICB. In the orthotopic model the effect on tumor onset is maybe not substantial as we focused on a clinically relevant setting and started treatment once tumors are about to become apparent (14 d after inoculation). In contrast, the effect is remarkably solid as with just three times of anti-PD-L1 treatment within 3 weeks the tumor volume was reduced to 1/2 and the effectiveness of ICB was confirmed by expected changes on immune cell markers. Similarly, in AOM/DSS tumors we do not observe response to ICB in Fibctrl mice, but a strong reduction in tumor growth, with some tumors even reducing their size, whereas others did not respond so well. This observation goes along with the relatively high variability in tumor size observed in this model upon *Zeb1* depletion.

We agree that among the many functions of TGF β in CRC, high TGF β signals are correlated with poor prognosis and refractoriness to ICB. We have shown that *Zeb1*-deficient CAFs are less responsive to TGF β and more prone to adopt and maintain an altered iCAF phenotype (Fig. 4). However, we did not analyze the TGF β abundance and modulation of TGF β would certainly also have consequences on other cell types in the TME. As *Zeb1* depletion in CAFs shifts the diversity of CAFs and alters their function to attract immune cells, which goes beyond the role of TGF β alone, we exclude a major TGF β -driven effect.

In short, I would recommend to go with your second option. Minimize or remove any claims on tumor growth, the combination with CPI and enthusiasm of therapeutic use (not that anyone will get excited by *Zeb1* as a therapeutic target at this time). Limit the paper to describing the change in polarization of CAFs and potential impact on immune contexture and go with EMBO Report rather than EMBO J.

We welcome that the arbitrator recommends our study for publication in EMBO Reports. We thoroughly went through the text and found a few instances where further toning down of conclusions deemed appropriate (labeled in red). We also explicitly addressed the "limitations of the study" in the last section of the discussion. Moreover, we now provide a graphical summary, in addition to minor text modifications to better explain unique features and the consistent findings in the different models.

Dear Dr. Stemmler,

Thank you for the submission of your further revised manuscript to our editorial offices. I now went through this and your p-b-p-response and consider the points of advisor #2 as adequately addressed.

Before proceeding with formal acceptance, I have these editorial requests I ask you to address in a final revised manuscript:

- Please provide a final title with not more than 100 characters (including spaces).
- We now use CRediT to specify the contributions of each author in the journal submission system. CRediT replaces the author contribution section. Please use the free text box to provide more detailed descriptions and do NOT provide your final manuscript text file with an author contributions section. See also our guide to authors: <https://www.embopress.org/page/journal/14693178/authorguide#authorshippinguidelines>
- Please make sure that the number "n" for how many independent experiments were performed, their nature (biological versus technical replicates), the bars and error bars (e.g. SEM, SD) and the test used to calculate p-values is indicated in the respective figure legends (main, EV and Appendix figures). Please also check that the exact p-values are indicated in the legend, and that these fit to those shown in the figure. Please provide statistical testing where applicable. Please avoid the phrase 'independent experiment', but clearly state if these were biological or technical replicates. Please also indicate (e.g. with n.s.) if testing was performed, but the differences are not significant. In case n=2, please show the data as separate datapoints without error bars and statistics. See also: <http://www.embopress.org/page/journal/14693178/authorguide#statisticalanalysis>

If n<5, please show single datapoints for diagrams. Presently, diagrams seem to miss the 'n.s.'. Please check.

Moreover:

- Please note that the exact p values are not provided in the legends of figures 1d, f-g, k-l; 2d; 3b, d-e, g-h; 4a-c, e-g, i; 5a-b, d; EV 1a; EV 3d; EV 4b-c, e; EV 5d-f.
- Please indicate the statistical test used for data analysis in the legends of figures 2d, i-j; EV 2c-d.
- Please note that in figures 4f; EV 4e; there is a mismatch between the annotated p values in the figure legend and the annotated p values in the figure file that should be corrected.
- Please note that information related to n is missing in the legend of figure 5d.
- Although 'n' is provided, please describe the nature of entity for 'n' in the legends of figures 3b; 4c; EV 1h, k; EV 3d.
- Please note that the error bars are not defined in the legend of figure 4f.
- Please add to each legend (main, EV and Appendix figures, where applicable) a 'Data Information' section explaining the statistics used or providing information regarding replicates and scales. See:

- Please add specific URLs that lead to the datasets GSE253368, GSE253639 and GSE253546 to the data availability statement, and make sure these datasets are public latest upon online publication of the manuscript.
- Co-corresponding author Thomas Brabletz is an EMBO reports board member. Please add the following sentence to the 'Disclosure and competing interests statement': "Thomas Brabletz is a member of the Advisory Editorial Board of EMBO reports. This has no bearing on the editorial consideration of this article for publication."
- There is a Fig. 2M called out in the manuscript text (page 7), but there seems to be no such panel. Please check.
- Please change the callouts for the Appendix Tables to 'Appendix Table Sx' (the "S" is presently missing).
- Thanks for providing the numerical source data (SD). You should have been contacted by our source data coordinator with information on which figure panels we would need source data for. I attach again the source data checklist. Please make sure that all the requested source data is provided. Please upload all source data for one figure ZIPed together as one folder. Please also upload the filled in source data checklist with your final revised files. Additional SD for EV and Appendix figures can be zipped up together in one folder.
- The schematic summary figure you have provided has not the right format. Scaling it down would render the text hardly readable. Please provide a final schematic summary figure as separate file that provides a sketch of the major findings (not a data image) in jpeg or tiff format with the exact width of 550 pixels and a height of not more than 400 pixels.

I look forward to seeing the final revised version of your manuscript when it is ready. Please let me know if you have questions

regarding the revision.

Please use this link to submit your revision: <https://embor.msubmit.net/cgi-bin/main.plex>

Best,

All editorial and formatting issues were resolved by the authors.

Dr. Marc Stemmler
Friederich-Alexander University of Erlangen-Nürnberg
Experimental Medicine 1
Glückstr. 6
Erlangen 91054
Germany

Dear Dr. Stemmler,

I am very pleased to accept your manuscript for publication in the next available issue of EMBO reports. Thank you for your contribution to our journal.

Yours sincerely,
